# SALUTARY LABELING WITH ZERO HUMAN ANNOTATION

## ABSTRACT

Active learning strategically selects informative unlabeled data points and queries their ground truth labels for model updates. The prevailing assumption in the active learning paradigm is that the acquisition of ground truth labels optimally enhances model performance. However, this assumption may not always hold or maximize learning capacity. Moreover, ground truth annotations incur significant costs due to the need for intensive human labor. In contrast to traditional active learning, this paper proposes salutary labeling, which automatically assigns the most beneficial labels to the most informative samples without human annotation. Specifically, we utilize the influence function, a tool for estimating sample influence, to select newly added samples and assign their salutary labels by choosing the category that maximizes their positive influence. This process eliminates the need for human annotation. Extensive experiments conducted on nine benchmark datasets demonstrate the superior performance of our salutary labeling approach over traditional active learning strategies. Additionally, we provide several in-depth explorations and practical applications including large language model fine-tuning.

## 1 INTRODUCTION

Active learning (Cohn et al., 1996; Zhan et al., 2022; Ren et al., 2021) is a specialized area in machine learning that focuses on effectively updating models by enabling them to request the labeling of particularly informative data points with a certain budget. This task arises from the challenge and expense involved in obtaining labeled data, which is often a major bottleneck in machine learning applications. To reduce labeling costs, active learning seeks to annotate only a small set of beneficial samples, which makes it particularly valuable when the labeling process is costly and time-consuming.

Consequently, significant research efforts have been dedicated to active learning in various research areas such as computer vision (Huang et al., 2018; Chai et al., 2021), natural language processing (Zhang et al., 2022; Ma et al., 2023), and medical diagnosis (Biswas et al., 2023; Wang et al., 2024). Traditionally, active learning methods select data points based on uncertainty and representativeness. The early uncertainty-based methods mainly measure the data uncertainty with the posterior probability predicted by the model (Holub et al., 2008; Wang et al., 2016; Balcan et al., 2007), while some recent approaches utilize auxiliary modules (Lakshminarayanan et al., 2017; Kee et al., 2018) to estimate uncertainty. Solely focusing on the uncertainty might cause bias in sampling, therefore other methods (Xu et al., 2003; Huang et al., 2018) aim to find the most representative subset of the full data. Recently, some studies (Liu et al., 2021; Chhabra et al., 2024) attempt to estimate the effect of integrating each data point on the training loss with the influence function (Cook & Weisberg, 1980).

The above active learning approaches show promising results but hinge on a critical assumption that training with ground truth labels of the selected samples will optimally enhance model performance. However, this assumption may not always hold, as some human-annotated labels can be incorrect or misleading, potentially harming the model's efficacy (Song et al., 2022; Chen et al., 2019). Moreover, even the correct label might harm or limit the model performance (Kong et al., 2021). Besides, the reliance on human-assigned labels in active learning inevitably incurs additional annotation costs.

**Contributions**. In this paper, we present salutary labeling, which aims to select the most informative samples and automatically annotate them with the most beneficial labels, enhancing training efficacy and eliminating the need for human intervention. We summarize our contribution as follows:

- We consider a new task named salutary labeling, which integrates the querying and annotating processes of active learning into a single autonomous step. To the best of our knowledge, this is the first initiative aimed at both maximizing model performance and eliminating the need for ground truth in active learning with an automatic labeling strategy.

- We adapt the influence function to calculate the sample influence, which serves as a criterion for selecting the most influential sample for labeling. However, the label information is required during calculating sample influence. Our salutary labeling ingeniously addresses this challenge by assessing the impact of each sample across all possible labels and assigning the label that yields the greatest positive influence. This simple strategy allows the model to automatically select and label samples, maximizing their overall benefit without any human annotation.

- We validate the efficacy of our approach on nine benchmark datasets, comparing with seven classical methods and two influence function-based methods in active learning. Beyond active learning experiments, we also conduct various in-depth explorations to address key questions for salutary labeling and extend its applications to other related tasks including LLM fine-tuning.

## 2 RELATED WORK

Our proposed salutary labeling introduces a new task that aims to query and annotate unlabeled samples in one unified step without any human intervention, which intersects with several areas within machine learning, particularly *active learning* (Cohn et al., 1996; Wei et al., 2015). Active learning selectively queries the user to annotate data points that are likely to be most beneficial for improving model performance, but contrasts with our method by relying on human annotations. Traditionally, some strategies (Zhan et al., 2022; Ren et al., 2021; Li et al., 2024) select important data points with indirect criteria such as uncertainty or representativeness. Uncertainty-based methods define sample uncertainty in one of three main ways: the entropy of the posterior probability distribution (Settles & Craven, 2008; Wu et al., 2022; Holub et al., 2008), the probability of the predicted class (Lewis & Catlett, 1994; Wang et al., 2016; Nguyen et al., 2022), or the margin between the probabilities of the highest two predicted classes (Joshi et al., 2009; Roth & Small, 2006; Balcan et al., 2007). Beyond these, research works (Freytag et al., 2014; Gal & Ghahramani, 2016) utilize consensus among multiple classifiers (Seung et al., 1992; Kee et al., 2018), or employ an auxiliary module (Yoo & Kweon, 2019) to measure uncertainty. Another strand of active learning approaches focuses on selecting the most representative samples (Xu et al., 2003; Huang et al., 2018; Sener & Savarese, 2018) through clustering (Nguyen & Smeulders, 2004) or by maximizing the distances between selected samples (Hasan & Roy-Chowdhury, 2015). Alternatively, several methods (Guo, 2010; Hasan & Roy-Chowdhury, 2015; Yang et al., 2015) attempt to identify the most diverse subset to represent the full dataset. Recently methods (Kirsch et al., 2019; Ash et al., 2020) effectively balance uncertainty and diversity by selecting data points that not only reduce model uncertainty but also ensure a diverse representation within each queried batch. Unlike these uncertainty-based and representativeness-based methods, our salutary labeling directly estimates each sample's impact on model performance with influence function.

Technically, our work is inherently related to *influence function* (Cook & Weisberg, 1980), which measures the change in a model's output due to an infinitesimal perturbation of one training data point. Following Koh & Liang (2017), significant research efforts (Giordano et al., 2019; Koh et al., 2019; Pruthi et al., 2020; Chen et al., 2021) are dedicated to quantifying the impact of individual or group of training samples on model performance. Recently, ISAL (Liu et al., 2021) extends the influence function to active learning by utilizing pseudo labels to calculate the influence. Alternatively, IBDS (Chhabra et al., 2024) incorporates an auxiliary regression module, which is specifically trained on labeled data and their calculated influences, to estimate the impact of unlabeled samples. While these methods avoid the requirement of labels in calculating influence function, they still rely on human annotators to label the selected data. In contrast, our method eliminates the need for human annotation, thereby avoiding the labor-intensive process of annotations and the potential inaccuracies associated with detrimental ground truth labels.

In terms of problem setting, our work is also related to *semi-supervised learning* (Yarowsky, 1995) and several data-centric topics (Hüllermeier & Beringer, 2005; Huggins et al., 2016; Kong et al., 2021; Li & Liu, 2022). We discuss these topics in detail in Appendix A due to space limitations.

## 3  MOTIVATION

Conventional active learning methods aim to strategically select unlabeled samples for annotation, assuming that correctly labeled samples inherently enhance model performance. However, this assumption may not always hold. Research in the realm of noisy labels (Natarajan et al., 2013; Song et al., 2022) has revealed that even a small subset of samples with noisy labels can contribute positively to model improvement. Our own observations, depicted in Figure 1 (top), further substantiate this claim. Leveraging the influence function, we discern the impact of individual samples on model performance. Based on this analysis, we calculate the sample influence with the most salutary label adjustment, maximizing its impact on model performance. Subsequently, we partition the entire training set into 20 equally-sized bins and replace the labels of samples within each bin with their optimal counterparts. Notably, the red line in the figure illustrates the model's performance with the entire training set, but with the labels of samples within each bin adjusted accordingly. Note that the dots representing equally-sized samples along the red line do not have uniform intervals and do not align with the unevenly-sized histogram. Surprisingly, for bins with high influence scores, retraining the model with these adjusted labels results in a significant performance improvement. For instance, in the last bin, the accuracy increases from 69% to 74%. This underscores the presence of salutary labels that surpass ground truth labels in enhancing model performance.

Expanding on the concept of salutary labels, we apply it within the framework of active learning, as depicted in Figure 1 (bottom). Analogous to our previous protocol, we sort the unlabeled samples based on their influence when labeled with salutary labels, dividing the unlabeled data into 20 equally-sized bins. The red and blue lines represent the performance when each bin is added to the labeled set with ground truth and salutary labels,

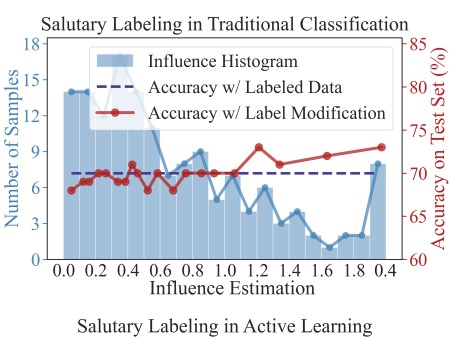

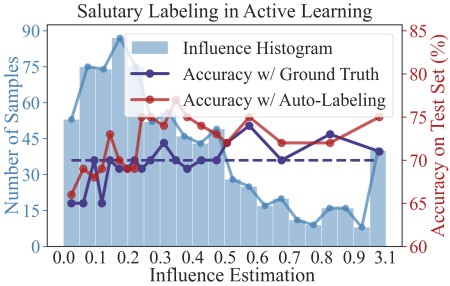

Figure 1: Experimental results on *Diabetes* (Decencière et al., 2014) dataset with ground truth and salutary labels. We select 300 labeled samples for traditional classification training and leave the remaining samples as unlabeled data from active learning. In both figures, the X-axis represents the sample influence with salutary labels. According to this measurement, we divide both labeled/unlabeled data into 20 equal-sized bins. The red and dark blue solid lines denote the performance of adding each bin into the labeled data with ground truth and salutary labels, respectively, and the dashed blue line denotes the performance when training with the original labeled data.

respectively. Our salutary labeling strategy consistently outperforms ground truth in most scenarios, particularly notable for samples with high influence estimations, which exhibit a remarkable 5% improvement over ground truth. It is noteworthy that the inclusion of bins with low influence leads to a decrease in accuracy, highlighting the presence of detrimental samples. These findings motivate us to pursue active learning with salutary labels, a strategy that not only enhances performance compared to ground truth but also alleviates the need for costly annotation effort.

## 4  METHOD

### 4.1  PRELIMINARIES

**Active learning**. The active learning process begins with training a model on a small initial labeled dataset $L=\{(x_i, y_i)\}_{i=1}^{N_L}$. Guided by certain criteria, active learning selects a small amount of the most informative unlabeled data points from a pool set $U=\{x_j\}_{j=1}^{N_U}$, queries their labels to obtain $B=\{(x_{j'}, y_{j'})\}_{j'=1}^{b}$, where $b$ represents the querying budget in each iteration, and updates the model with the newly labeled data $L \cup B$. These queried samples are then removed from the unlabeled pool for subsequent iterations. This learning cycle is repeated for multiple rounds, gradually enhancing model performance while minimizing labeling effort.

**Influence function**. For a labeled training dataset $\{(x_i, y_i)\}_{i=1}^N$ and a model with a convex loss function $\ell(\cdot, \cdot)$, the optimized parameters for empirical risk minimization can be represented as $\hat{\theta} = \arg\min_{\theta \in \Theta} \frac{1}{N} \sum_i \ell(x_i, y_i) + \frac{\lambda}{2} \|\theta\|_2^2$. If one training point $(x_j, y_j)$ is downweighted by infinitesimal $\epsilon$ during the training, the new optimized parameters change to $\hat{\theta}_{(x_j, y_j); -\epsilon} = \arg\min_{\theta \in \Theta} \frac{1}{N} \sum_i \ell(x_i, y_i) - \epsilon \ell(x_j, y_j) + \frac{\lambda}{2} \|\theta\|_2^2$. Without actually retraining the model, the influence function (Cook & Weisberg, 1980) estimates the actual change by $\hat{\theta}_{(x_j, y_j); -\epsilon} - \hat{\theta} = -\mathbf{H}_{\hat{\theta}}^{-1} \nabla_{\hat{\theta}} \ell(x_j, y_j)$, where $\mathbf{H}_{\hat{\theta}} = \frac{1}{N} \sum_{i=1} \nabla_{\hat{\theta}}^2 \ell(x_i, y_i) + \lambda \mathbf{I}$ is the Hessian matrix for $\hat{\theta}$.

By setting $\epsilon = 1/N$, we can linearly approximate the change of $\hat{\theta}$ after removing a training sample, as removing sample $(x_j, y_j)$ is equivalent to down-weighting it with $\epsilon = 1/N$. If the validation set $V$ is taken into consideration, let the validation loss be $\mathcal{L}_v = \ell(V; \hat{\theta})$, the impact of a specific training data point $(x_j, y_j)$ on the validation loss can be estimated as follows (Koh & Liang, 2017):

$$\mathcal{I}(x_j, y_j) = -\nabla_{\hat{\theta}} \mathcal{L}_v^\top \mathbf{H}_{\hat{\theta}}^{-1} \nabla_{\hat{\theta}} \ell(x_j, y_j). \tag{1}$$

Unlike traditional active learning methods that rely on indirect criteria such as uncertainty (Balcan et al., 2007; Yang et al., 2015; Nguyen et al., 2022) or representativeness (Huang et al., 2010; Du et al., 2015; Gu et al., 2021) to select informative samples, the influence function offers a more direct and precise assessment of a data point's importance to the model. By quantifying the effect of each sample on the model loss on the validation set, the influence function provides a more accurate means of selecting the most informative data points for labeling. Despite the potential benefits, the influence function presents a crucial challenge in active learning. As shown in Eq. (1), the influence function relies on having label information to estimate the impact of each data point, which poses a challenge when dealing with pool samples in the active learning task where such labels are unavailable. Previous influence-based methods use pseudo-labels or surrogate models to avoid directly addressing this challenge. Instead, our approach introduces salutary labeling to overcome this obstacle, which is a simple and effective labeling strategy and makes the influence function flexible for active learning.

## 4.2 SALUTARY LABELING FOR ACTIVE LEARNING

In this work, we propose salutary labeling for active learning, a novel approach that directly evaluates the impact of each unlabeled sample and automatically assigns labels to the selected data without any human annotation. Our method fulfills the requirement for ground truth labels in influence function calculation, by systematically exploring all possible labels for each data point and calculating the influence corresponding to each label. The label with the highest influence estimation is then assigned to each sample as the salutary label. This salutary influence, estimated using the salutary label, represents the maximum possible benefit when incorporating the data point into training. Subsequently, our method selects the unlabeled samples with the highest salutary influence and annotates them with salutary labels in a unified step, without requiring any human intervention. In the following section, we introduce the notations and provide technical details of our method.

**Training protocol and technical notations**. In each iteration of active learning, the model is trained on the labeled training set $L$ with label space $\mathcal{C}$. The optimized model parameters for the convex training loss function $\ell(\cdot, \cdot)$ are denoted as $\hat{\theta}$. To actively query the most beneficial samples from the unlabeled pool set $U = \{x_i\}_{i=1}^{N_U}$, our salutary labeling algorithm calculates the influence estimation of every data point $x_i$ with its salutary label on the validation loss $\mathcal{L}_v = \ell(V; \hat{\theta})$. The samples with the highest influences are selected as the salutary set, denoted as $B = \{(x_j, y_j^s)\}_{j=1}^b$, where $y_j^s \in \mathcal{C}$ represents the salutary label of the queried data and the superscript 's' represents the salutary label. After forming the salutary set, it is removed from the pool $U$, thus updating $U = U \setminus B$. Subsequently, the model is re-trained on the expanded labeled set $L = L \cup B$ for the next active learning cycle.

**Salutary labeling with the influence function**. With the concept of the salutary label, we can handle the absence of label information when calculating the influence function. Specifically, for an unlabeled sample, we compute the influence estimations for each label and pick the one with the largest influence, ensuring the most beneficial label is chosen. Mathematically, it can be expressed as:

$$\mathcal{I}(x_j, y_j^s) = \mathcal{I}(x_j, \hat{c}), \text{ where } \hat{c} = \arg\max_{c \in \mathcal{C}} \mathcal{I}(x_j, c). \tag{2}$$

**Autonomous active learning**. Eq. (2) directly measures the impact of each unlabeled sample and automatically assigns the salutary label, enabling our method to query and annotate the unlabeled data without human intervention. Specifically, the model selects the top $b$ samples with the highest influences from the pool set $U$ and annotates them with salutary labels, to form an active salutary set $B = \{(x_j, y_j^s)\}_{j=1}^b$. This salutary set is then removed from unlabeled set $U$ and integrated into the labeled training set $L$, to update the learning model.

We summarize the training protocol of salutary labeling in Algorithm 1 of the Appendix. The time complexity of salutary labeling is bounded by the calculation of the influence function in Eq. (2). For each label $c \in \mathcal{C}$, the calculation of gradients for all unlabeled samples will take $\mathcal{O}(nd)$, where $n$ is the number of samples and $d$ is the dimension of model parameter $\theta$. Notice that the computation of the Hessian matrix and its inverse only involves the label information of the validation set. Therefore, these calculations only need to be performed once for all potential labels. The explicit computation of Hessian takes $\mathcal{O}(nd^2)$ and its inversion takes $\mathcal{O}(d^3)$. We apply conjugate gradients and stochastic estimations of Hessian-vector products (Koh & Liang, 2017), reducing the time complexity to $\mathcal{O}(nd)$.

## 5 EXPERIMENTS

In this section, we first introduce our experimental setup, then report the algorithmic performance of extended active learning experiments, and finally provide in-depth analyses of salutary labeling.

### 5.1 EXPERIMENTAL SETUP

**Datasets and baseline methods**. We use six tabulate datasets from UCI Machine Learning Repository (Dua et al., 2017) in our experiments. We also use the 39 pre-extracted features of *CelebA* (Liu et al., 2015) as a tabulate dataset. Additionally, we include two vision dataset, *MNIST* (Deng, 2012) and *CIFAR10* (Krizhevsky & Hinton, 2009). We use a ResNet-34 (He et al., 2016), which is pre-trained on the ImageNet (Deng et al., 2009), to extract 512 deep features for each image in both datasets. We provide details of each dataset in Appendix B.

We include the nine baseline methods for active learning. Random sampling is the most intuitive baseline which randomly queries samples from the pool set. Entropy sampling (Holub et al., 2008) selects the unlabeled samples with the highest entropy of the current model's predictions. Margin sampling (Balcan et al., 2007) ranks all pool samples by the margin between the highest and second-highest values from the soft-max logits predicted by the model. Uncertainty sampling (Nguyen et al., 2022) queries by the classification uncertainty, which is determined by the probability of the predicted class as assigned by the classifier. CoreSet (Sener & Savarese, 2018) focuses on selecting the most representative and diverse subset of the data to query for labeling. BatchBALD (Kirsch et al., 2019) utilizes Bayesian principles to maximize the expected reduction in uncertainty over a batch by considering the mutual information of selected instances. BADGE (Ash et al., 2020) selects points based on their expected information gain while maintaining diversity within each batch. We also include two influence-based active learning methods, which choose the unlabeled data set with influence estimation. ISLA (Liu et al., 2021) uses base model predictions as pseudo-labels to compute influence. IBDS (Chhabra et al., 2024) uses an influence regressor, which is trained with labeled training data and their influences calculated with Eq. (1), to predict the influence estimations for the unlabeled data. It is important to note that while all baseline methods require human effort to annotate the queried unlabeled samples, our approach is completely human annotation-free.

**Implementation details and experimental protocol**. We implement[1] our method with Scikit-learn (Pedregosa et al., 2011) and Pytorch (Paszke et al., 2019). All experiments are conducted on our workstation equipped with one 24GB NVIDIA TITAN RTX GPU. In our experiments, we divide all datasets into training set (60%), validation set (20%), and test set (20%), except for *Bank*, *CelebA* and *Diabetic* datasets, which have predefined splits for training, validation, and testing. The influence-based models, including ISAL, IBDS, and our methods, exclusively utilize the validation set to compute influence estimations. This setup ensures that none of the methods access any information from the test set, maintaining the testing data unseen to the models during the evaluation. All experiments are repeated five times with different random seeds. In each run, we randomly choose 300 samples from the training set as the initial set and reserve the rest as the pool set.

---

[1]Our code is available at *https://anonymous.4open.science/r/salutary-labeling-11CF*.

Table 1: Accuracy (%) of the logistic regression model on the test set after 10 rounds of active learning in five runs. We report only the average accuracy in this table due to the space limits. The standard deviations are presented in Figure 2 as well as in Table 4 in Appendix C.

| Method | Electric | Bank | Diabetic | CelebA | Musk_v2 | Wine | Waveform | CIFAR10 | MNIST |
|---|---|---|---|---|---|---|---|---|---|
| Init | 63.85 | 65.89 | 56.43 | 73.33 | 73.45 | 44.76 | 79.11 | 46.74 | 77.75 |
| Random | 65.15 | 67.77 | 58.41 | 82.06 | 78.33 | 46.31 | 81.10 | 55.92 | 80.93 |
| Entropy 2008 | 69.72 | 73.84 | 65.34 | 81.23 | 79.11 | 45.00 | 83.23 | 53.91 | 83.77 |
| Margin 2007 | 69.72 | 73.84 | 65.34 | 81.23 | 79.11 | 47.30 | 82.26 | 56.95 | 83.72 |
| Uncertainty 2022 | 69.72 | 73.84 | 65.34 | 81.23 | 79.11 | 44.53 | 83.33 | 55.47 | 83.63 |
| ISLA 2021 | 67.98 | 64.41 | 61.38 | 84.71 | 77.72 | 47.15 | 79.40 | 53.91 | 79.35 |
| IBDS 2024 | 67.66 | 65.14 | 64.35 | 82.49 | 78.15 | 44.84 | 82.91 | 54.61 | 80.05 |
| CoreSet 2018 | 66.35 | 68.21 | 61.38 | 80.14 | 73.78 | 47.61 | 80.70 | 54.64 | 81.26 |
| BatchBALD 2019 | 67.06 | 74.15 | 64.76 | 78.85 | 77.53 | 46.69 | 81.83 | 53.66 | 82.05 |
| BADGE 2020 | 67.45 | 74.92 | 64.16 | 81.19 | 78.48 | 46.87 | 81.21 | 56.44 | 84.24 |
| Ours w/ GT | 70.92 | 66.45 | 68.31 | 83.03 | 77.34 | 48.23 | 83.74 | 55.92 | 86.12 |
| Ours w/ SL | **71.31** | **78.07** | **71.28** | **85.50** | **81.06** | **49.92** | **84.21** | **58.33** | **86.68** |
| Diff. GT *vs.* SL | 14 | 19 | 13 | 10 | 22 | 7 | 8 | 11 | 8 |

We choose a logistic regression classification model that satisfies the convex requirement of the influence function. We initiate the process by training this model with the initial set. Subsequently, we conduct active learning for $R = 10$ active rounds. In each round, the model queries 10 samples from the pool dataset $U$. For baseline methods, the ground truth labels of these selected samples are used, whereas our method automatically assigns salutary labels according to Eq. (2). After labeling, the queried data points are integrated into the labeled set for re-training the model. After each round of learning, we evaluate the model's performance by measuring prediction accuracy on the test set.

We set the query budget $b$ to 10 to maintain the distinction in performance between different models. Using a larger budget, such as 1% of the pool set, might cause the model to reach the performance ceiling on some datasets. We provide a detailed discussion and visualization on this in Appendix D.

## 5.2 ALGORITHMIC PERFORMANCE

We evaluate the performance of our salutary labeling method alongside the active learning baselines. Note that the entropy, margin, and uncertainty samples yield the same results for the same random initial/pool splits in binary classification datasets, as these three metrics have the same rank for 2-dimensional logits. We add our method with Ground Truth (GT) as a baseline, where the same unlabeled samples queried by our method are annotated with ground truth labels for comparison. We also compared the differences between the salutary labels (SL) and the ground truth labels, counting how many of the 100 queried samples have discrepancies between the two sets of labels.

As shown in Table 1, our method shows significant improvements over the initial model despite a limited querying budget and achieves the highest accuracy among all active learning methods. We notice that the two influence-based baselines do not perform well on datasets like *Diabetic* and *Wine*. This highlights the difficulties in estimating influence without access to label information, emphasizing the challenges and limitations of current influence-based approaches in handling complex datasets where salutary labeling shows a clear advantage. Our method with ground truth labels achieves promising results, and salutary labeling further improves the accuracy across all datasets. Notably, salutary labeling differs from ground truth in only a limited number of samples, as in the last row of Table 1. The performance boost from this small set of different labels validates that salutary labeling can identify key instances and assign more beneficial labels based on the validation set.

Moreover, we also present the accuracy change over 10 learning rounds for all methods in Figure 2. Our method shows significant and steady improvements, particularly in challenging datasets like *Bank*, *Waveform*, and *Wine*, where the baselines show limited progress. This indicates the efficiency of salutary labeling in active learning, particularly noteworthy as it requires no human annotations.

In addition to logistic regression, we also conduct active learning experiments for ResNet-34 (He et al., 2016) model on *CIFRA10* and *MNIST* and achieve promising performance. We report the detailed results in Appendix E. These results validate the ability of our method to perform autonomously across different models and datasets, highlighting its potential for practical applications.

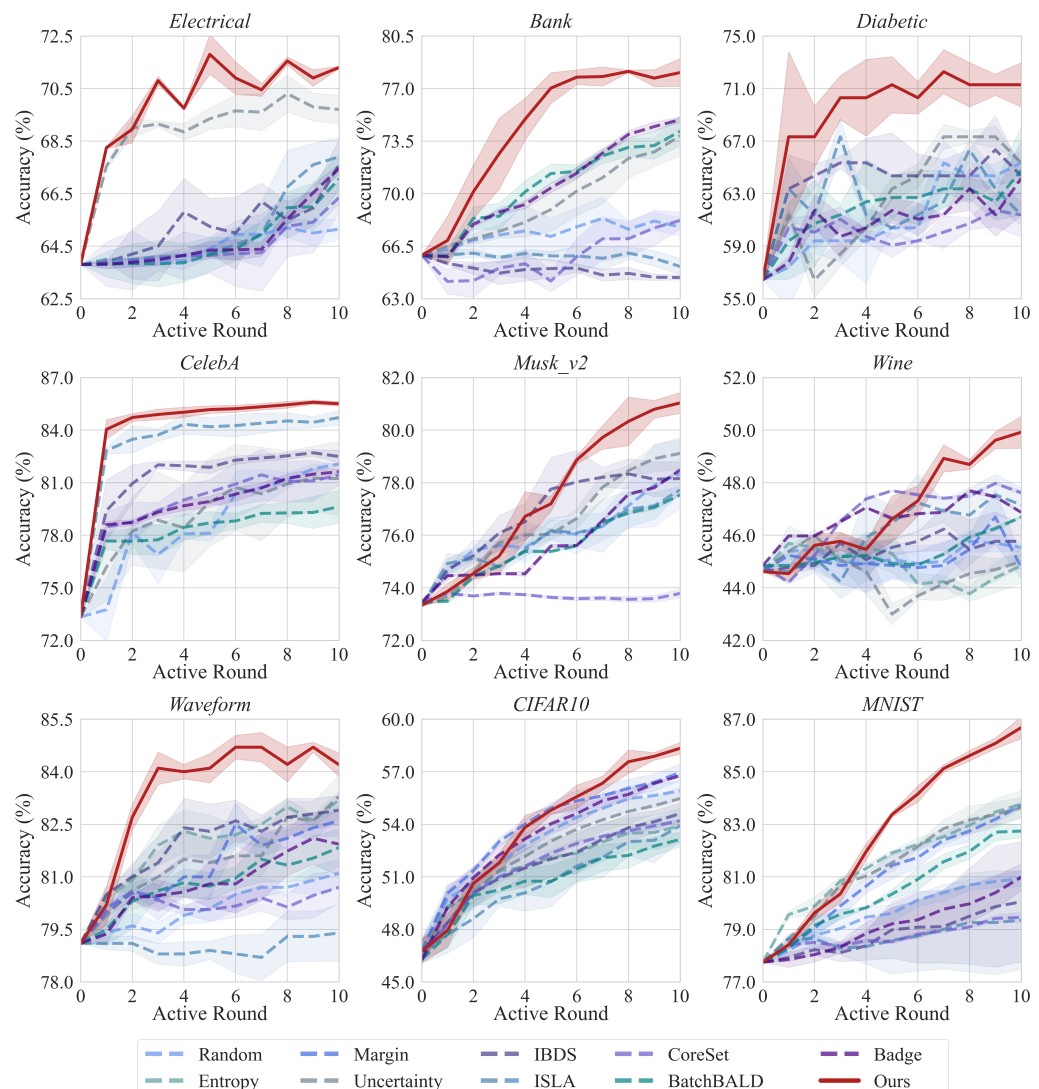

Figure 2: Comparison of salutary labeling and baseline active learning methods on nine datasets over 10 rounds of learning cycle. In all figures, the X-axis represents the training iterations, where round 0 is the initial training. The shaded area is the standard deviation across 5 different random runs. Notice that the entropy, margin and uncertainty sampling yield the same results for binary datasets.

## 5.3 IN-DEPTH EXPLORATIONS

We would like to answer the following questions for salutary labeling in our in-depth explorations:

- The influence function has been demonstrated as an accurate estimation for leave-one-out influence (Koh & Liang, 2017), which estimates the impact of removing a training sample. On the contrary, salutary labeling adapts this function to assess the effect of adding a sample unseen during model training, raising the question: How accurate is this estimation?

- As salutary labeling does not require human annotation, there is no budget constraint. Is it possible to achieve better performance when training with more pool samples?

- The influence function requires the learning model to be convex, which limits its applied scenarios. Can we circumvent the convex requirement of influence function and extend the salutary labeling to applications involving non-convex deep models?

**Influence estimation vs. add-one-in retraining**. We empirically verify how accurate is the influence function when estimating the impact of adding a new data point on three datasets, namely *Diabetic*, *CelebA*, and *Bank*. For each dataset, we compare the predicted influence estimations with the actual

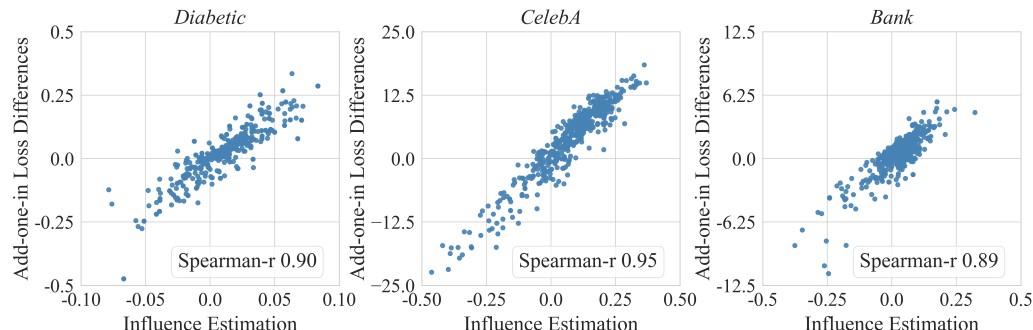

Figure 3: Influence estimation vs. actual loss difference of add-one-in retraining on *Diabetic* (left), *CelebA* (middle), and *Bank* (right) datasets. In all plots, the horizontal axes represent the estimated influence on validation loss, while the vertical axes show the actual loss change. Their correlation is quantified with the Spearman's rank correlation coefficient (Spearman-r). We randomly selected 300 samples in each plot to ensure clarity in visualization.

changes in loss observed after adding a sample and re-training the model. Using the initial set, we train a logistic regression model $\hat{\theta}$ and compute the influence $\mathcal{I}(x_j, y_j)$ for every data point in the pool set. Consequently, we individually add each pool sample $(x_j, y_j)$ to the training set and update the model parameters $\hat{\theta}_j$. We compare influence estimation $\mathcal{I}(x_j, y_j)$ and the validation loss difference after add a sample $\ell(V; \hat{\theta}_j) - \ell(V; \hat{\theta})$. As shown in Figure 3, The influence estimation for new samples does not perfectly match the actual loss change, likely because they were unseen during initial training. Still, the influence estimations are highly correlated with actual loss differences, as measured by Spearman's rank correlation coefficient. Therefore, the influence function provides an accurate indication of each sample's relative impact.

**Salutary labeling with more data points**. In Section 5.2, we demonstrated the efficacy of salutary labeling. The fact that salutary labeling requires zero human intervention allows our method to query even more unlabeled samples without incurring any annotation costs. Therefore, we conduct additional experiments to evaluate the effectiveness of our method with more pool samples. Following the setup described in Section 5.2, we split the data into an initial set for training the initial logistic regression model, along with a pool set, validation set, and test set. For each data set, the model queries and automatically annotates 10 samples from the pool set with salutary labeling in each active learning iteration. We allow the model to query up to 50% samples from the pool set and choose the iteration that has the best predicting accuracy on the validation set as the final model.

In addition to evaluating our salutary labeling, we report the test accuracy obtained after training the model with all labeled data from both the initial and pool sets. This provides a reference point to the maximum achievable accuracy when the model is supervised by all available data. We also include two semi-supervised learning (SSL) methods, namely, self-training (Yarowsky, 1995) and FixMatch (Sohn et al., 2020), as they similarly leverage a small labeled dataset alongside a larger pool of unlabeled data to enhance model performance.

As demonstrated in Table 2, our method consistently outperforms the SSL baselines across all datasets, showing that salutary labeling benefits from utilizing the validation set. Moreover, our method achieves higher accuracy than supervised learning on four datasets, validating that salutary labels can provide superior guidance compared to ground truth labels under certain conditions. On *Musk_v2* (Chapman & Jain, 1994), *Wine* (Cortez et al., 2009), and *Waveform* (Breiman & Stone, 1988) datasets, the fully supervised model only leads our method by a narrow margin of less than 1%. On *CIFAR10* (Krizhevsky & Hinton, 2009) and *MNIST* (Deng, 2012), our method trails the fully supervised model by about 3.5%, but it still boosts the accuracy by over 15% compared to the initial model. Notably, these gains are achieved without any human annotation, illustrating the effectiveness of our salutary labeling in utilizing unlabeled data.

**Salutary labeling for LLM fine-tuning**. In this section, we aim to expand our salutary labeling method to practical applications with complex model structures. Specifically, we conduct the active learning experiments in the LLM fine-tuning with a non-convex RoBERTa (Liu et al., 2019) model on three datasets of *GLUE* (Wang et al., 2018) repository, namely, *WNLI* (Levesque et al., 2011),

Table 2: Accuracy (%) of the logistic regression model on the test set after querying 50% of the pool set. The standard deviations (less than 0.3% for all datasets) are omitted due to space limits.

| Method | Electrical | Bank | Diabetic | CelebA | Musk_v2 | Wine | Waveform | CIFAR10 | MNIST |
|---|---|---|---|---|---|---|---|---|---|
| Initial | 63.85 | 65.89 | 56.43 | 73.33 | 73.45 | 44.76 | 79.11 | 46.74 | 77.75 |
| Fully Supervised | 70.08 | 80.14 | 72.27 | 85.07 | **85.75** | **52.53** | **85.60** | **65.67** | **95.36** |
| Self-Training 1995 | 64.85 | 72.22 | 59.4 | 77.07 | 74.48 | 46.84 | 83.50 | 47.24 | 77.86 |
| FixMatch 2020 | 66.47 | 73.83 | 60.14 | 76.82 | 76.52 | 47.65 | 82.85 | 51.85 | 82.38 |
| Ours | **72.25** | **81.21** | **73.26** | **85.89** | 85.68 | 52.38 | 85.50 | 62.05 | 92.06 |

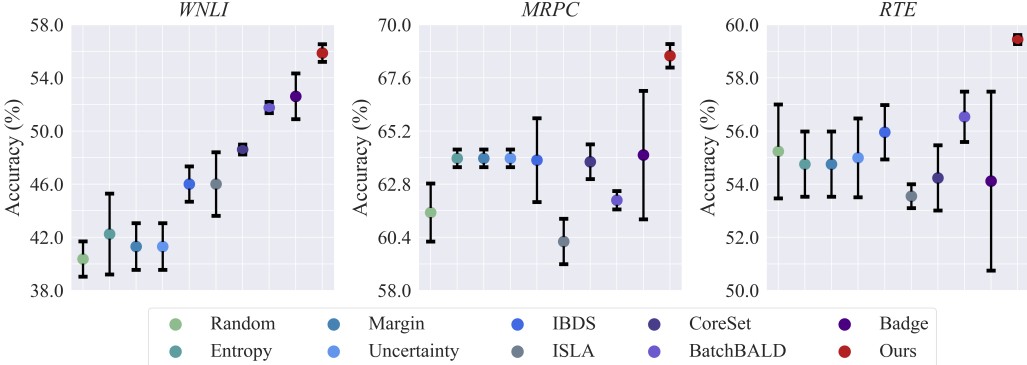

Figure 4: Accuracy of the final model after 10 rounds of active learning for LLM fine-tuning on *WNLI* (left), *MRPC* (middle) and *RTE* (right) datasets of *GLUE* repository.

*MRPC* (Dolan & Brockett, 2005) and *RTE* (Bentivogli et al., 2017). We simulate an active learning scenario for fine-tuning the RoBERTa model, denoted by $g \circ h$, where $g$ represents the transformer layers and $h$ represents the classification head. Following the setting of Section 5.1, we divide each dataset into the initial set, pool set, validation set, and test set.

During the whole training, we fix the transformer layers $g$ in RoBERTa and fine-tune the non-convex classification head $h$. Initially, we train the model using the initial set. Subsequently, in each learning cycle, we use the 768-dimensional hidden state extracted by $g$, along with predictions from $h$, to train a surrogate logistic regression model $h'(\cdot; \hat{\theta})$. This surrogate model was then used to identify and annotate 10 samples from the pool set, as detailed in Algorithm 1. The newly annotated samples are used to update the classification head $h$. We provide the training details in Appendix F.

As illustrated in Figure 4, our method outperforms all baseline approaches in all three tasks after 10 learning cycles. The performance advantage is consistent across most rounds, with detailed per-round results displayed in Figure 6 of the Appendix F. These findings underscore the potential of our method in practical applications, highlighting the adaptability and effectiveness of our approach in real-world settings, even when the model is not strictly convex.

## 6 CONCLUSION

In this paper, we delved into the realm of active learning and proposed a novel concept called salutary labeling, which seamlessly merges the querying and annotating processes of active learning into a single autonomous step. Unlike traditional methods, our approach eliminates the need for human annotation; instead, it automatically assigns a salutary label, i.e., the label category that maximizes model performance. Technically distinct from conventional active learning approaches that rely on indirect measurements such as uncertainty and representativeness to select samples for labeling, we utilized the influence function to directly compute sample influence. However, a significant challenge arises when dealing with pool samples in active learning tasks, as label information may be unavailable. Our salutary labeling method adeptly overcomes this hurdle by evaluating the impact of each sample across all possible labels and assigning the label that generates the greatest positive influence. Extensive experimental results underscored the efficacy and advantages of our salutary labeling approach across various scenarios.

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

APPENDIX

## A    RELATED WORKS

**Semi-supervised learning**. Semi-supervised learning (SSL) leverages both labeled and unlabeled data to improve model performance. Early semi-supervised learning methods like self-training (Yarowsky, 1995) and co-training (Blum & Mitchell, 1998) use iterative self-labeling and multi-view learning to exploit unlabeled data. Pseudo-labeling (Lee, 2013) extends this idea by assigning high-confidence predictions to unlabeled examples, though it encountered challenges related to label noise. Consistency regularization techniques, such as Pi-models (Laine & Aila, 2022) and VAT (Miyato et al., 2019), address this by enforcing prediction stability under data perturbations. Ensemble methods like Mean Teacher (Tarvainen & Valpola, 2017) improved SSL by refining predictions through a stable teacher model. Recent advances, such as FixMatch (Sohn et al., 2020; Zhang et al., 2021) and FlexMatch (Zhang et al., 2021) combine strong data augmentation with pseudo-labeling, further enhancing SSL performance by enforcing consistency between weak and strong augmentations.

**Other data-centric topics**. *Data relabeling* methods (Yang & Yu, 2023; Kong et al., 2021) seek to relabel the harmful training samples for better model performance, while *partial label learning* (Hüllermeier & Beringer, 2005; Lyu et al., 2020; Gong et al., 2022) aims to train a classifier to accurately predict the ground-truth label using partially labeled data, where each training instance is associated with multiple candidate labels. Although both tasks involve automatically assigning labels to data points, neither of them is designed to query unseen samples for further improving model performance. *Data-efficient learning* (Huggins et al., 2016; Munteanu et al., 2018; Coleman et al., 2019; Paul et al., 2021) aims to accelerate model training by selecting a minimum subset of the data, which requires ground truth labels for all available data. *Antidote data* (Li & Liu, 2022; Rastegarpanah et al., 2019) overlaps with our method as it generates additional training data to modify specific model behaviors such as fairness or robustness. However, these data-centric approaches do not primarily focus on the active learning task.

## B    DATASETS

We use the seven tabulate datasets and two vision datasets in our experiments. *Bank* (Moro et al., 2014) dataset has a total of 30,488 records of bank telemarketing phone calls. Each sample contains 51 features which are used to predict if a client will subscribe to a term deposit or not. *Diabetic* (Decencière et al., 2014) dataset contains 1,151 retina images of patients for predicting if the patients suffer from Diabetes or not. We use 19 features extracted by Antal & Hajdu (2014). *CelebA* (Liu et al., 2015) has a total of 104,163 samples of face images with 39 features from each sample image and we treat the features as tabulated data to predict if the person is smiling or not. *Musk_v2* (Chapman & Jain, 1994) dataset contains 6,598 instances of molecules, and 166 features to represent the low-energy conformations of the molecules, which is used to learn to predict whether new molecules will be musks or non-musks. *Electrical* (Arzamasov, 2018) dataset contains 10,000 points and 11 attributes such as power consumption and price in a 4-node star electrical grid system, which is used to predict if the system is stable or not. *Wine* (Cortez et al., 2009) dataset consists of the physicochemical test results for 4,898 variants of the Portuguese "Vinho Verde" wine. We use it to predict the quality scores (from 3 to 9) based on 11 physicochemical attributes, such as acidity, density, and alcohol rate. *Waveform* (Breiman & Stone, 1988) dataset contains 5,000 instances of waveform records, each described by 21 attributes. We use it to classify each record into one of the three waveform classes. *MNIST* (Deng, 2012) is a collection of 70,000 handwritten digit images (0 through 9). We use a ResNet-34 (He et al., 2016), which is pre-trained on the ImageNet (Deng et al., 2009), to extract 512 deep features for each image. *CIFAR10* (Krizhevsky & Hinton, 2009) consists of 60,000 real-life images in 10 classes, with 6,000 images per class. Similar to the *MNIST* dataset, we also extract 512 features with the pre-trained ResNet-34.

We summarize some key statistics of the nine datasets we use in Section 5.2 in Table 3. For all datasets, we conduct five runs of experiments with different random seeds. In each run, we fix the validation set and test set, then randomly choose 300 samples from the training samples as the initial set, and reserve the rest as the pool set. All datasets are publicly available with *CC BY 4.0* license.

Table 3: Dataset Statistics

| Dataset | # of Train | # of Val | # of Test | # of Classes | # of Dim | Data Type |
|---------|-----------|----------|-----------|--------------|----------|-----------|
| *Bank* 2014 | 18,292 | 6,098 | 6,098 | 2 | 51 | tabulate |
| *Diabetic* 2014 | 950 | 100 | 100 | 2 | 19 | tabulate |
| *CelebA* 2015 | 62,497 | 20,833 | 20,833 | 2 | 39 | tabulate |
| *Musk_v2* 1994 | 3,958 | 1,320 | 1,320 | 2 | 166 | tabulate |
| *Electrical* 2018 | 6,000 | 2,000 | 2,000 | 2 | 12 | tabulate |
| *Waveform* 1988 | 3,000 | 1,000 | 1,000 | 3 | 21 | tabulate |
| *Wine* 2009 | 3,896 | 1,300 | 1,300 | 7 | 11 | tabulate |
| *MNIST* 2012 | 54,000 | 6,000 | 10,000 | 10 | 512 | vision |
| *CIFAR10* 2009 | 45,000 | 5,000 | 10,000 | 10 | 512 | vision |

## C  DETAILED ALGORITHMIC PERFORMANCE WITH STANDARD DEVIATION

We do not include the standard deviation in Table 1 for better visualization. Here we report the full experimental results with standard deviation in Table 4, which includes the active learning experiments in Section 5.2 and the LLM fine-tuning experiments in Section 5.3. Our salutary labeling method outperforms all baseline methods across multiple datasets in the standard active learning setting for both convex logistic regression and non-convex LLM fine-tuning, all without requiring any human annotation. Notably, our method not only achieves the highest final predicting accuracy across all datasets but also maintains relatively small standard deviations, keeping consistent performance across different experimental runs. These results highlight the efficacy of our method, emphasizing its potential in practical applications.

Table 4: Accuracy (%) for all datasets on the test data after 10 learning cycles, alongside the standard deviations across five experimental runs with different random seeds. This table includes the results of all experimental in Section 5.2 and LLM fine-tuning in Section 5.3.

| Method | *Electrical* | *Bank* | *Diabetic* | *CelebA* | *Musk_v2* | *Wine* |
|--------|-----------|--------|-----------|----------|-----------|--------|
| Init | 63.85 | 65.89 | 56.43 | 73.33 | 73.45 | 44.76 |
| Random | $65.15_{\pm0.40}$ | $67.77_{\pm0.61}$ | $58.41_{\pm0.93}$ | $82.06_{\pm0.13}$ | $78.33_{\pm1.30}$ | $46.31_{\pm0.16}$ |
| Entropy 2008 | $69.72_{\pm0.55}$ | $73.84_{\pm1.33}$ | $65.34_{\pm0.11}$ | $81.23_{\pm2.11}$ | $79.11_{\pm0.60}$ | $45.00_{\pm0.41}$ |
| Margin 2007 | $69.72_{\pm0.55}$ | $73.84_{\pm1.33}$ | $65.34_{\pm0.11}$ | $81.23_{\pm2.11}$ | $79.11_{\pm0.60}$ | $47.30_{\pm0.25}$ |
| Uncertainty 2022 | $69.72_{\pm0.55}$ | $73.84_{\pm1.33}$ | $65.34_{\pm0.11}$ | $81.23_{\pm2.11}$ | $79.11_{\pm0.60}$ | $44.53_{\pm0.67}$ |
| ISLA 2021 | $67.98_{\pm0.74}$ | $64.41_{\pm0.54}$ | $61.38_{\pm0.80}$ | $84.71_{\pm0.41}$ | $77.72_{\pm0.14}$ | $47.15_{\pm0.65}$ |
| IBDS 2024 | $67.66_{\pm0.94}$ | $65.14_{\pm0.15}$ | $64.35_{\pm0.46}$ | $82.49_{\pm0.36}$ | $78.15_{\pm0.64}$ | $44.84_{\pm0.64}$ |
| CoreSet 2018 | $66.35_{\pm0.56}$ | $68.21_{\pm0.68}$ | $61.38_{\pm0.15}$ | $80.14_{\pm0.52}$ | $73.78_{\pm0.14}$ | $47.61_{\pm0.09}$ |
| BatchBALD 2019 | $67.06_{\pm0.94}$ | $74.15_{\pm0.20}$ | $64.76_{\pm0.33}$ | $78.85_{\pm0.18}$ | $77.53_{\pm0.06}$ | $46.69_{\pm0.06}$ |
| BADGE 2020 | $67.45_{\pm0.16}$ | $74.92_{\pm0.86}$ | $64.16_{\pm0.28}$ | $81.19_{\pm0.89}$ | $78.48_{\pm0.09}$ | $46.87_{\pm0.04}$ |
| Ours | $\mathbf{71.31}_{\pm0.04}$ | $\mathbf{78.07}_{\pm0.92}$ | $\mathbf{71.28}_{\pm1.68}$ | $\mathbf{85.50}_{\pm0.12}$ | $\mathbf{81.06}_{\pm0.39}$ | $\mathbf{49.92}_{\pm0.61}$ |

| Method | *Waveform* | *CIFAR10* | *MNIST* | *WNLI* | *MRPC* | *RTE* |
|--------|-----------|-----------|---------|--------|--------|-------|
| Init | 79.11 | 46.74 | 77.75 | 40.69 | 60.13 | 52.87 |
| Random | $81.10_{\pm0.39}$ | $55.92_{\pm0.52}$ | $80.93_{\pm0.30}$ | $40.77_{\pm1.33}$ | $61.51_{\pm1.31}$ | $55.23_{\pm1.76}$ |
| Entropy 2008 | $83.23_{\pm0.44}$ | $53.91_{\pm0.46}$ | $83.77_{\pm0.13}$ | $42.25_{\pm3.04}$ | $63.95_{\pm0.40}$ | $54.73_{\pm1.19}$ |
| Margin 2007 | $82.26_{\pm0.59}$ | $56.95_{\pm0.64}$ | $83.72_{\pm0.38}$ | $41.32_{\pm1.75}$ | $63.89_{\pm0.38}$ | $54.78_{\pm1.24}$ |
| Uncertainty 2022 | $83.33_{\pm0.23}$ | $55.47_{\pm1.01}$ | $83.63_{\pm0.50}$ | $41.31_{\pm1.64}$ | $63.93_{\pm0.42}$ | $54.99_{\pm1.48}$ |
| ISLA 2021 | $79.40_{\pm0.80}$ | $53.91_{\pm0.87}$ | $79.35_{\pm1.87}$ | $46.01_{\pm2.39}$ | $60.21_{\pm0.45}$ | $53.54_{\pm1.02}$ |
| IBDS 2024 | $82.91_{\pm0.41}$ | $54.61_{\pm0.60}$ | $80.05_{\pm2.28}$ | $45.98_{\pm1.32}$ | $63.88_{\pm1.02}$ | $55.95_{\pm1.02}$ |
| CoreSet 2018 | $80.70_{\pm0.45}$ | $45.21_{\pm0.18}$ | $79.45_{\pm0.19}$ | $48.61_{\pm0.07}$ | $63.81_{\pm0.14}$ | $54.23_{\pm0.12}$ |
| BatchBALD 2019 | $81.94_{\pm0.09}$ | $53.13_{\pm0.41}$ | $81.00_{\pm0.26}$ | $52.61_{\pm0.17}$ | $64.11_{\pm0.29}$ | $54.21_{\pm0.33}$ |
| BADGE 2020 | $81.83_{\pm0.06}$ | $56.77_{\pm0.10}$ | $82.74_{\pm0.25}$ | $51.77_{\pm0.05}$ | $62.08_{\pm0.09}$ | $56.53_{\pm0.09}$ |
| Ours | $\mathbf{84.21}_{\pm0.40}$ | $\mathbf{58.33}_{\pm0.33}$ | $\mathbf{86.68}_{\pm0.42}$ | $\mathbf{55.86}_{\pm0.66}$ | $\mathbf{68.59}_{\pm0.53}$ | $\mathbf{59.44}_{\pm0.17}$ |

## D  CHOICE OF QUERY BUDGET $b$

We set a relatively small query budget $b$ to maintain clear performance distinctions between different models. In our preliminary exploration stage, we found that a larger budget, such as 1% of the

pool set, allows models to reach the performance ceiling on datasets like *CelebA* (Liu et al., 2015), *Waveform* (Breiman & Stone, 1988), and *Electrical* (Arzamasov, 2018). As shown in Figure 5, such a budget causes different active learning methods to perform very similarly after several rounds. Consequently, we opted for a smaller budget in our experiments to better evaluate the distinct capabilities of each model.

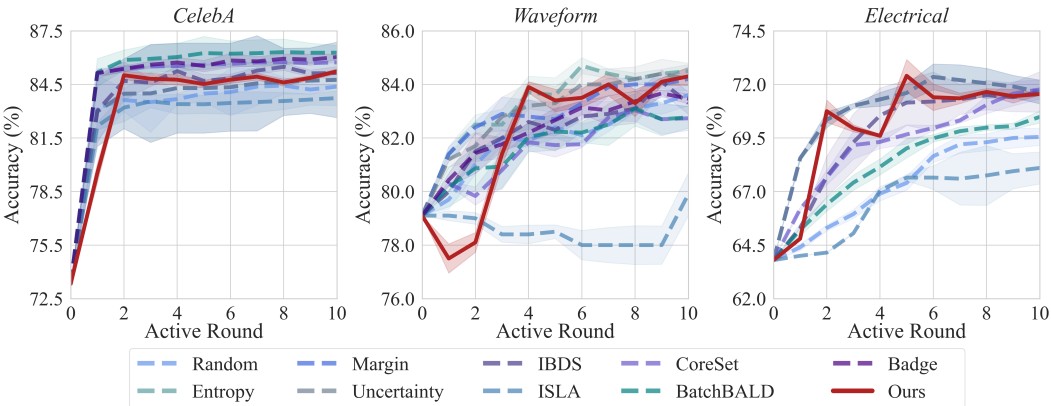

Figure 5: Prediction accuracy of salutary labeling and baseline methods with $b$ set at 1% of the pool samples on *CelebA* (left), *Waveform* (middle), and *Electrical* (right) datasets.

## E  EXPERIMENTAL RESULTS FOR RE-TRAINING THE NEURAL NETWORK

We also conduct experiments with deep ResNet-34 (He et al., 2016) on raw images of *MNIST* (Deng, 2012) and *CIRAR10* (Krizhevsky & Hinton, 2009), where we re-train the full neural network after acquiring additional annotations. Following the experimental protocol outlined in Section 5.2, we start with training the ResNet-34 model with 300 labeled samples and query 10 unlabeled samples in each of the total 10 learning rounds. For influence-based methods, we use a surrogate logistic regression model to calculate the influence function. This surrogate model is trained on the 512-dimensional representations extracted by the ResNet-34 and the predicted labels from its classification layer. Our method still outperforms the active learning baselines by a small margin, further demonstrating the potential of salutary labeling in practical settings, even with non-convex models.

We also notice that training the full network achieves worse accuracy than only training the logistic regression model with extracted representations. This can happen because a very

Table 5: Accuracy (%) for *CIFAR10* and *MNIST* datasets on ResNet-34 after re-training the full model for 10 active learning cycles.

| Method | CIFAR10 | MNIST |
|---|---|---|
| Init | 11.8 | 15.68 |
| Random | 25.09 | 58.95 |
| Entropy 2008 | 39.15 | 61.42 |
| Margin 2007 | 40.25 | 62.89 |
| Uncertainty 2022 | 40.25 | 63.04 |
| ISLA 2021 | 36.64 | 64.41 |
| IBDS 2024 | 40.13 | 62.64 |
| CoreSet 2018 | 39.27 | 58.53 |
| BatchBALD 2019 | 40.16 | 64.64 |
| BADGE 2020 | 39.95 | 65.13 |
| Ground Truth (GT) | 40.02 | 65.45 |
| Ours | **41.61** | **66.32** |

small training set is insufficient for training deep neural networks due to the risk of overfitting and the inability to generalize effectively to unseen data (Shorten & Khoshgoftaar, 2019). Deep networks thrive on vast and diverse data, as their numerous parameters need large datasets to capture complex patterns. In contrast, using a logistic regression model on pre-trained, fixed representations reduces the risk of overfitting, as simpler models require fewer parameters to train and utilize the extracted features more efficiently (Kornblith et al., 2019).

## F  TRAINING DETAILS FOR ACTIVE LLM FINE-TUNING

We conduct our LLM fine-tuning experiments on three datasets of GLUE (Wang et al., 2018) repository, namely, *WNLI* (Levesque et al., 2011), *MRPC* (Dolan & Brockett, 2005) and *RTE* (Bentivogli et al., 2017). *WNLI* is a reading comprehension dataset, where the authors construct sentence pairs by replacing the ambiguous pronoun in the original sentence with each possible referent. This dataset is used for predicting whether the sentence with the pronoun substituted is entailed by the original

sentence or not. *MRPC* is a corpus of sentence pairs automatically extracted from online news sources, and we use it to predict whether the sentences in the pair are semantically equivalent or not. *RTE* are constructed based on news and Wikipedia text. The task is to classify each sample into one of the two classes assigned by human annotators.

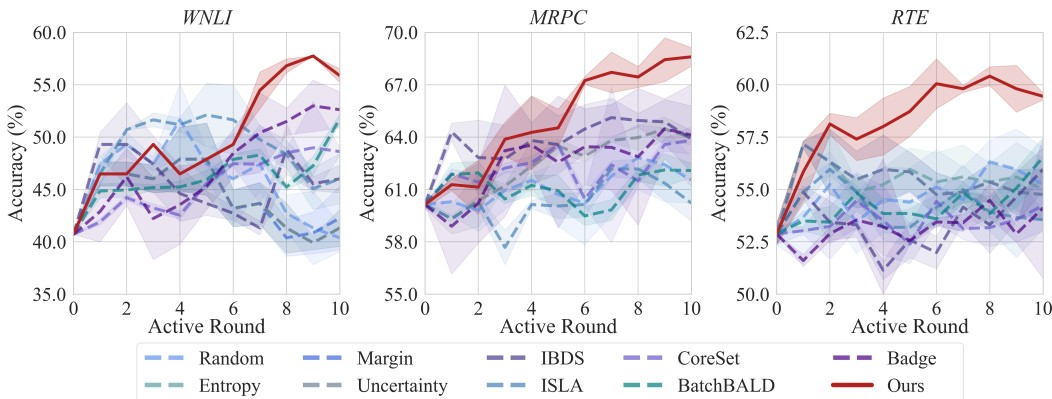

Figure 6: Comparison of salutary labeling and baseline methods on three datasets of GLUE (Wang et al., 2018) repository over 10 rounds of learning cycle in LLM fine-tuning application.

For each dataset, we randomly select 100 samples from the predefined training split to form the initial set and use the remaining data as the pool set. Half of the predefined validation split serves as the validation set for salutary labeling, with the other half used as the test set. We use the Hugging Face (Wolf et al., 2020) implementation of RoBERTa (Liu et al., 2019), denoted by $g \circ h$. We fix the transformer layers $g$ while fine-tuning the classification head $h$, which is a two-layer multilayer perceptron model with dropout before both layers and $tahn$ activation function between the two layers. Initially, the model is trained with the initial set. In each of the 10 active learning cycles, it annotates 10 samples. For sampling methods like entropy, margin, and uncertainty, the output of $h$ determines the pool set queries. For influence-based methods including ISAL (Liu et al., 2021), IBDS (Chhabra et al., 2024) and our method, we train a surrogate logistic regression model $h'(\cdot; \hat{\theta})$ using the 768-dimensional hidden states extracted by $g$ and predictions from $h$. This surrogate model was then used to calculate the influence function and query pool samples for model re-training. We compute the accuracy on the test set after each round and plot the results in Figure 6.

## G   BROADER IMPACT AND LIMITATIONS

This paper presents work whose goal is to advance the field of machine learning. We broaden the scope of active learning with a novel approach called salutary labeling, which integrates the querying and annotating processes of active learning into a single, autonomous step. The proposed salutary labeling method eliminates human annotation and maximizes benefits from queried data. Beyond the impact mentioned above, there are also other potential societal consequences of our work, none of which we feel must be specifically highlighted here.

One potential limitation of our method stems from the influence function, one key component of salutary labeling. The influence function requires the model to be convex, ensuring that its Hessian matrix is positive definite, and invertible after training to convergence. Despite the ongoing discussions (Basu et al., 2020; Bae et al., 2022; Epifano et al., 2023) on the accuracy of the influence function on non-convex models, many research works have successfully applied the influence function across various applications (Fang et al., 2020; Han et al., 2020; Chen et al., 2023). In this work, we adopt the same strategy as in the work of Li & Liu (2022), which uses a surrogate convex model on the embeddings extracted by the non-convex model, and achieve promising results as illustrated in Section 5.3. Further exploring the application of the influence function to non-convex models is not the focus of this study, so we defer this topic to future work.

## H  CODE AND REPRODUCIBILITY

The code for the implementation of our method will be publicly available in the following repository: *https://anonymous.4open.science/r/salutary-labeling-11CF*.

All experiments were conducted on a Linux workstation running Ubuntu 20.04.6 LTS. The CPU used was an Intel(R) Core(TM) i9-10850K CPU @ 3.60GHz. Any experiments requiring GPU (such as multi-class influence calculation and LLM fine-tuning) were conducted with one NVIDIA TITAN RTX GPU with 24 VRAM and CUDA version 11.4.

All codes are written in Python, and utilize basic libraries such as NumPy (Harris et al., 2020), scikit-learn (Pedregosa et al., 2011), PyTorch (Paszke et al., 2019), Pandas (Wes McKinney, 2010), etc. Detailed package information will be provided in the code repository.

## ALGORITHM

---

**Algorithm 1** Salutary Labeling for Active Learning

---

**Input:** Labeled training set $L$, unlabeled pool set $U$, validation set $V$ and model parameters $\theta$.
**Parameters:** Total active training round $R$ and the query budget $b$.

1: Train the model and obtain the optimized parameters $\hat{\theta}$ with loss term $\frac{1}{N_L}\Sigma_{(x_i,y_i)\in L}\ell(x_i,y_i)$.
2: **for** $r = 1$ to $R$ **do**
3:   **for** $x_j \in U$ **do**
4:     Calculate the sample influence with its salutary label $\mathcal{I}(x_j, y_j^s)$ by Eq. (2).
5:   **end for**
6:   Select $b$ samples with the highest influence as salutary set $B = \{(x_j, y_j^s)\}_{j=1}^b$.
7:   Update the labeled training set as $L = L \cup B$.
8:   Remove the salutary set from the pool set as $U = U \setminus B$.
9:   Re-train the model with $L$ and update $\hat{\theta}$.
10: **end for**

**Output:** The final optimized model parameters $\hat{\theta}$ after $R$ rounds of active learning.

---

