# OpenReview forum: "Salutary Labeling with Zero Human Annotation"
_ICLR.cc/2025/Conference — Submitted to ICLR 2025_

### Official Review · Reviewer_qHRp · 2024-10-27

**Soundness:** 3
**Presentation:** 3
**Contribution:** 3
**Rating:** 6
**Confidence:** 4

**Summary:**

The authors propose salutary labeling for active learning which is human annotation-free. They adapt the influence function to calculate the sample influence by assessing the impact of each sample across all possible labels and assigning the label that yields the greatest positive influence. The authors conducted experiments on different datasets to verify the effectiveness of the simple idea.

**Strengths:**

1.The writing is good and the supplementary materials are relatively sufficient.

2.The authors propose a simple-sounding but effective active learning method which eliminates the need for human annotation. Judging from the comparative experimental results provided by the authors, this idea is effective.

**Weaknesses:**

1.In the last paragraph of Section 4, the authors mentioned that the time complexity of salutary labeling is O(nd). However, the proposed salutary labeling algorithm need to calculate the influence estimation of every data point in each iteration of active learning. How much will this slow down the entire training process? Can the authors provide the running time comparison results of each method in Table 1?

2.In the experiment, the author set active rounds R = 10 and query budget b = 10. When b and R are larger, is it impossible to prove that the proposed salutary labeling is effective?

3.The first paragraph of Section 2 is too long and a little bit difficult to read. It should be adjusted appropriately.

4.The legend in Figure 1 obscures part of the polyline and may need to be further refined.

**Questions:**

1.In the last paragraph of Section 4, the authors mentioned that the time complexity of salutary labeling is O(nd). However, the proposed salutary labeling algorithm need to calculate the influence estimation of every data point in each iteration of active learning. How much will this slow down the entire training process? Can the authors provide the running time comparison results of each method in Table 1?

2.In the experiment, the author set active rounds R = 10 and query budget b = 10. When b and R are larger, is it impossible to prove that the proposed salutary labeling is effective?

3.The first paragraph of Section 2 is too long and a little bit difficult to read. It should be adjusted appropriately.

4.The legend in Figure 1 obscures part of the polyline and may need to be further refined.

---

> ### Author Response · Authors · 2024-11-19
> **Response to Reviewer qHRp (1/2)**
>
> Thank Reviewer qHRp's kind efforts in reviewing our paper. We are happy to see the reviewer's recognition of the beauty of simplicity inherent in our approach. We would like to provide
> point-to-point responses as follows.
>
> **Running time**. Great suggestion! Here we listed the average running for 10 active learning rounds in one experimental run in the following table. As shown in Table A2, our method achieves shorter running times on datasets with 2 or 3 categories. However, for CIFAR10 and MNIST, it runs slower than the baselines due to the significant increase in the number of classes, which proportionally raises the number of parameters in the model (*d*). Despite this, we emphasize that our method eliminates the need for human annotations, which can potentially save substantial time in real-world applications.
>
> Additionally, various research efforts [1, 2, 3] have focused on reducing the time required for influence calculations. These methods have achieved promising results, enabling efficient influence estimation for large datasets and complex models. As optimizing the computation of influence functions lies outside the scope of this work, we defer the acceleration of influence function calculations to future research.
>
> ***Table A2***: Running time (in seconds) of 10 rounds of active learning for different methods.
> | **Method**              | **Electric** | **Bank** | **Diabetic** | **CelebA** | **Musk_v2** | **Wine** | **Waveform** | **CIFAR10** | **MNIST** |
> |-------------------------|:------------:|:--------:|:------------:|:----------:|:-----------:|:--------:|:------------:|:-----------:|:---------:|
> | *Random*              | 2.1          | 1.6      | 1.0          | 4.3        | 12.4        | 5.6      | 2.0          | 22.4        | 23.8      |
> | *Entropy*            | 2.4          | 2.3      | 1.2          | 4.8        | 13.6        | 5.8      | 2.5          | 24.2        | 26.6      |
> | *Margin*              | 2.4          | 2.3      | 1.2          | 4.8        | 13.6        | 5.8      | 2.5          | 24.2        | 26.7      |
> | *Uncertainty*         | 2.4          | 2.3      | 1.2          | 4.8        | 13.6        | 5.8      | 2.5          | 24.1        | 23.6      |
> | *CoreSet*             | 12.8         | 5.1      | 1.8          | 40.7       | 19.1        | 6.9      | 4.9          | 120.4       | 129.4     |
> | *BatchBALD*           | 11.8         | 7.5      | 1.5          | 59.5       | 21.8        | 8.9      | 5.5          | 143.6       | 131.6     |
> | *BADGE*               | 10.4         | 8.9      | 2.1          | 48.6       | 19.7        | 12.6     | 5.6          | 181.4       | 144.0     |
> | *ISLA*                | 4.2          | 8.8      | 2.2          | 33.4       | 16.8        | 14.1     | 8.1          | 410.2       | 364.0     |
> | *IBDS*                | 2.9          | 4.7      | 1.8          | 9.7        | 14.5        | 7.5      | 4.2          | 189.6       | 151.5     |
> | *Ours*                | 4.2      | 8.8  | 2.2      | 33.4   | 16.9   | 14.1 | 8.1      | 410.5  | 386.2 |
>
> [1] Pruthi, Garima, et al. "Estimating training data influence by tracing gradient descent." NeurIPS 2020.
>
> [2] Kwon, Yongchan, et al. "DataInf: Efficiently Estimating Data Influence in LoRA-tuned LLMs and Diffusion Models." ICLR. 2023.
>
> [3] Grosse, Roger, et al. "Studying large language model generalization with influence functions." arXiv preprint arXiv:2308.03296. 2023.
>
> We will continue our discussion in the comment ***section (2/2)***, due to the space limit.

---

> ### Author Response · Authors · 2024-11-19
> **Response to Reviewer qHRp (2/2)**
>
> **Larger query rounds *R* or larger query budget *b***. We provide our rationale for selecting a smaller query budget *b* in Appendix D, where we demonstrate that a larger budget causes the classification performance to reach a ceiling on the datasets. Regarding additional rounds, we invite Reviewer qHRp to refer to the second question in Section 5.3, where we conduct more learning rounds and annotate 50\% of the pool set with salutary labels. For the convenience of the reviewer, we include the relevant results here. As shown in Table 2, our method achieves comparable or even higher performance than fully supervised training on the entire pool data, highlighting its effectiveness over more rounds of active learning.
>
> ***Table***: Accuracy (%) of the logistic regression model on the test set after querying 50% of the pool set.
> | Method               | Electrical |  Bank   | Diabetic | CelebA  | Musk_v2 |  Wine   | Waveform | CIFAR10 |  MNIST  |
> |----------------------|:----------:|:-------:|:--------:|:-------:|:-------:|:-------:|:--------:|:-------:|:-------:|
> | *Initial*          |   63.85    |  65.89  |   56.43  |  73.33  |  73.45  |  44.76  |   79.11  |  46.74  |  77.75  |
> | *Fully Supervised* |   70.08    |  80.14  |   72.27  |  85.07  | **85.75**| **52.53**| **85.60**| **65.67**| **95.36**|
> | *Ours*            | **72.25**  | **81.21**| **73.26**| **85.89**|  85.68  |  52.38  |   85.50  |  62.05  |  92.06  |
>
> **Presentation suggestions**. Thank the reviewer for the insightful suggestions for improving our presentation. We will split the first paragraph of Section 2 to make it easier to follow and understand. We will also make refinements in Figure 1 to highlight the polylines.
>
> If there is anything in our response remains unclear to the reviewer, we are happy to provide more evidence and explanations.

---

### Official Review · Reviewer_otRr · 2024-10-29

**Soundness:** 2
**Presentation:** 4
**Contribution:** 2
**Rating:** 5
**Confidence:** 4

**Summary:**

This study defines a task called salutary labeling, which involves selecting a subset of data from an unlabeled data pool predicted to be the most beneficial for training, similar to traditional active learning, and then using this selected data for model training. The key difference in the proposed salutary labeling approach is that, instead of using labels from human annotators, it assigns pseudo labels expected to improve performance on the validation set. Specifically, it measures the influence of each data point and assigns a salutary label that maximizes validation set performance. The data with the highest influence from the assigned salutary labels is then used for training. This method was tested on nine datasets and demonstrated improved results compared to traditional active learning approaches.

**Strengths:**

- The *Salutary Labeling for Active Learning* framework was new to me. If we can effectively annotate data automatically without human annotator, it could demonstrate remarkable potential for machine learning as a whole.

- Overall, the writing is well-structured and easy to follow, making it straightforward to understand the main concepts.

- The proposed method consistently improved performance across nine datasets. This study also conducted detailed ablation studies to analyze the effectiveness of *salutary labels* through the influence function.

**Weaknesses:**

- The motivation behind combining salutary labeling with active learning is not fully clear to me.
   - The core motivation of active learning is to label only a small number of informative data points to reduce annotation costs. If automatic labeling without human annotation costs is feasible, applying salutary labels to all available data without the selection process in active learning should suffice.
   - In Algorithm 1, it seems that salutary labels are generated for the entire unlabeled pool during selection. Figure 5 also suggests that model performance improves as the number of salutary labels increases, so the necessity of sampling is unclear.
   - **Additional explanation on why the active learning framework remains effective despite not requiring human labor for labeling would help clarify my understanding.**

- I am not fully convinced about how the salutary labeling task is novel compared to existing label-efficient tasks.
   - To me, the salutary labeling task appears to combine elements from various existing label-efficient tasks. If the goal is to automatically label training data, this could be viewed as a pseudo-labeling process in semi-supervised or self-supervised learning. On the other hand, methods for correcting label noise align more closely with the learning with label noise task [a, b, c].
   - In Appendix A, a key difference from other label-efficient learning tasks is described as a focus on active learning (line 836). However, as mentioned above, without a clear reason for combining salutary labels with the active learning framework, I remain uncertain about the task’s distinction from others.
   - **A clearer explanation on why salutary labels need to be combined with active learning and how this approach fundamentally differs from other label-efficient learning methods would be helpful.**

- There seem to be potential fairness concerns in the comparative experiments with existing AL baselines.
   - The proposed experimental setup uses 20% of the dataset as a validation set and utilizes this validation set’s annotations for influence function calculations and labeling. If labeled data is essential for the proposed method, it would be fairer to allocate part of the training budget for this purpose.
   - Specifically, in Table 1, the budget sample size is set as low as 10, which makes it impractical to reserve 20% of the dataset as a validation set while only training on about 10 samples. This setup might give the impression that validation set labels are indirectly being used for salutary labeling of the unlabeled data.
   - As shown in Figure 5 of Appendix D, one potential reason for the reduced gain of the proposed method as the budget size increases may be the dilution of the validation set’s supervisory effect.
   - **The following additional experiments might strengthen the justification for salutary labeling:**
       - Using a smaller validation set (e.g., 1% to 5% instead of 20%).
       - Including the validation set size in the training budget for all methods (e.g., allowing AL baselines to train on the validation set or at least part of it).

[a] Xiao, Ruixuan, et al. "Promix: Combating label noise via maximizing clean sample utility." arXiv preprint arXiv:2207.10276 (2022).

[b] Chen, Wenkai, Chuang Zhu, and Mengting Li. "Sample prior guided robust model learning to suppress noisy labels." Joint European Conference on Machine Learning and Knowledge Discovery in Databases. Cham: Springer Nature Switzerland, 2023.

[c] Liu, Sheng, et al. "Robust training under label noise by over-parameterization." International Conference on Machine Learning. PMLR, 2022.

**Questions:**

- The proposed approach appears closely related to training with noisy labeled data methods mentioned in the Weakness section. An analysis and comparison with this literature in the related work would be beneficial.
- A detailed analysis of the assigned salutary labels would be interesting. It would be helpful to know if labels different from the ground truth sometimes enhance performance or if performance improvement primarily comes from refining noisy labels.

---

> ### Author Response · Authors · 2024-11-19
> **Response to Reviewer otRr (1/4)**
>
> Thank you for your kind efforts in reviewing our paper. We are happy to see the reviewer's informative comments. We would like to provide point-to-point responses as follows.
>
> **Why not applying salutary labels to all available**. We follow the active learning settings, where data is selected and labeled iteratively rather than all at once.
> This iterative process enhances the accuracy of the influence function calculation by dynamically updating the model with newly labeled data points.
> Instead of relying on a fixed model, we continuously update the model after each annotation round, recalculating the influence estimation based on the updated model.
> This allows for more targeted and precise labeling, ensuring that each new annotation is maximally beneficial to the model.
> This dynamic adjustment enhances the effectiveness of salutary labels and optimally improves the model performance.
>
> **No human annotation**. We agree with the reviewer that human-assigned ground truth labels intuitively seem to provide performance boost in active learning. Yet we believe that learning models inherently exhibit biases that do not always align with the human perspective. This misalignment can result in scenarios where ground truth labels, though accurate from a human perspective, fail to provide the most benefit to the model's learning process, causing the ground truth labels to yield suboptimal model performance. Notably, previous works on noisy labels [1,2] have demonstrated that incorporating a small number of labels considered noisy by humans can improve model performance. Our own observations, detailed in Figure 1 and Table 1 of the manuscript (See the table below), further support this argument, showing that selectively leveraging such "noisy" but salutary labels can enhance learning outcomes for the model.
>
> Our method takes this perspective of the model, enables the active learning framework to identify the most impactful samples, and assigns their corresponding salutary labels.
> By prioritizing data that maximizes performance gains, this approach optimizes the model's learning potential without relying on human annotation efforts.
>
>
> ***Table***: Accuracy (%) of the logistic regression model on the test set after 10 rounds of active learning of salutary labeling and ground truth. The last row represents the number of different annotations assigned by salutary labels and ground truth labels out of 100 new annotations.
> | Method                  | **Electric** | **Bank** | **Diabetic** | **CelebA** | **Musk_v2** | **Wine** | **Waveform** | **CIFAR10** | **MNIST** |
> |-------------------------|:------------:|:--------:|:------------:|:----------:|:-----------:|:--------:|:------------:|:-----------:|:---------:|
> | *Init*                    | 63.85        | 65.89    | 56.43        | 73.33      | 73.45       | 44.76    | 79.11        | 46.74       | 77.75     |
> | *Ours w/ GT*              | 70.92        | 66.45    | 68.31        | 83.03      | 77.34       | 48.23    | 83.74        | 55.92       | 86.12     |
> | *Ours w/ SL*              | **71.31**    | **78.07**| **71.28**    | **85.50**  | **81.06**   | **49.92**| **84.21**    | **58.33**   | **86.68** |
> |                         |              |          |              |            |             |          |              |             |           |
> | *Diff. GT vs. SL*     | **14**       | **19**   | **13**       | **10**     | **22**      | **7**    | **8**        | **11**      | **8**     |
>
>
> [1] Natarajan, Nagarajan, et al. "Learning with noisy labels." NeurIPS. 2013.
>
> [2] Song, Hwanjun, et al. "Learning from noisy labels with deep neural networks: A survey." IEEE TNNLS. 2022.
>
> We will continue our discussion in the comment ***section (2/4)***, due to the space limit.

---

> ### Author Response · Authors · 2024-11-19
> **Response to Reviewer otRr (2/4)**
>
> **Novelty of our work compared with other label-efficient methods**. Our salutary labeling can be regarded as a novel labeling method with broad applicability across various tasks, particularly in active learning and semi-supervised learning. Here we would like to clarify that we chose to showcase its effectiveness mainly within the active learning context in this work for two main reasons: i) salutary labeling demonstrates a marked improvement in model performance when applied to active learning, underscoring its value in this area; ii) By generating high-quality salutary labels, our method significantly reduces the need for human annotations. Here, we would like to point out that our novelty comes from our insight into the research problem and the fresh perspective we bring to addressing its challenges. We humbly believe this constitutes a key novelty of a research paper.
>
> Specifically, our method shifts away from a human-centric perspective of existing label-efficient methods in active learning or noisy label tasks. Instead, our work focuses entirely on the model's viewpoint in annotating data, which aims to identify the data most beneficial for the model. Following this model-driven perspective, we extend the influence function to the context of active learning, enabling the model to directly estimate the impact of each unlabeled data point and its associated salutary label on model performance. This enables our method to address two fundamental challenges of active learning: estimating the impact of data points and reducing human annotation costs. We believe this new perspective fundamentally differs from other related methods in active learning or noisy label tasks. Technically, our method differs from [a, b, c] as it evaluates unseen data without requiring annotations and does not focus on the "correctness" of labels.
>
> We will continue our discussion in the comment ***section (3/4)***, due to the space limit.

---

> ### Author Response · Authors · 2024-11-19
> **Response to Reviewer otRr (3/4)**
>
> **Fair comparison regarding validation set**.
> Thanks to the reviewer for the suggestion to explore different sizes of validation sets.
>
> i) We conducted the experiments for smaller validation sets. We randomly sampled validation sets as 10%, 5% and 1% of the data, and conducted the experiments for all baselines. We left out the *Diabetic* dataset as it is ready a small dataset its validation set only contains 100 samples. For traditional methods without the influence functions, i.e., the baselines other than ISLA and IBDS, the performance of different-sized validation sets is quite small (with 1% accuracy), so we reported the overall average accuracy here.
>
>  Our performance remains robust for validation sets comprising 10% and 5% of the data as shown in the table. While we observe a performance drop with only 1\% validation data, our method still slightly outperforms other baselines, except on the *Electric*, *Bank*, and *CIFAR10* datasets, as highlighted with underlines. This suggests that the size of the validation set does not significantly impact our method's effectiveness, as long as it provides a relatively accurate direction for influence estimation.
>
>  However, using very small validation sets might pose a limitation, as they may not provide sufficient information to maximize the guidance required for accurate influence estimation. Despite this, our method achieves comparable performance without any reliance on human annotations, demonstrating its robustness even under constrained validation settings.
>
> ***Table A1***: Accuracy (%) of the logistic regression model on the test set after 10 rounds of active learning with different validation sizes.
>
> | Method              | Electric |  Bank   | CelebA  | Musk_v2 |  Wine   | Waveform | CIFAR10 |  MNIST  |
> |---------------------|:--------:|:-------:|:-------:|:-------:|:-------:|:--------:|:-------:|:-------:|
> | *Init*            |  63.85   |  65.89  |  73.33  |  73.45  |  44.76  |  79.11   |  46.74  |  77.75  |
> | *Random*          |  65.15   |  67.77  |  82.06  |  78.33  |  46.31  |  81.10   |  55.92  |  80.93  |
> | *Entropy*         |  69.72   |  73.84  |  81.23  |  79.11  |  45.00  |  83.23   |  53.91  |  83.77  |
> | *Margin*          |  69.72   |  73.84  |  81.23  |  79.11  |  47.30  |  82.26   |  56.95  |  83.72  |
> | *Uncertainty*     |  69.72   |  73.84  |  81.23  |  79.11  |  44.53  |  83.33   |  55.47  |  83.63  |
> | *CoreSet*         |  66.35   |  68.21  |  80.14  |  73.78  |  47.61  |  80.70   |  54.64  |  81.26  |
> | *BatchBALD*       |  67.06   |  74.15  |  78.85  |  77.53  |  46.69  |  81.83   |  53.66  |  82.05  |
> | *BADGE*           |  67.45   |  74.92  |  81.19  |  78.48  |  46.87  |  81.21   |  56.44  |  84.24  |
> |-------------------------|------------|------------|------------|------------|------------|------------|------------|------------|
> | *ISLA* w/ ***10% Val*** |  67.98   |  64.41  |  84.71  |  77.72  |  47.15  |  79.40   |  53.91  |  79.35  |
> | *IBDS* w/ ***10% Val*** |  67.66   |  65.14  |  82.49  |  78.15  |  44.84  |  82.73   |  54.82  |  80.05  |
> | *Ours*  w/ ***10% Val*** | **71.31**| **78.07**|**85.50**|  81.06  | **49.92**|  84.21   | **58.33**|**86.68**|
> |-------------------------|------------|------------|------------|------------|------------|------------|------------|------------|
> | *ISLA* w/ ***5% Val***  |  64.41   |  68.61  |  84.28  |  79.46  |  44.62  |  79.40   |  50.54  |  79.41  |
> | *IBDS* w/ ***5% Val***  |  67.65   |  65.52  |  81.91  |  77.51  |  45.38  |  81.86   |  54.21  |  78.34  |
> | *Ours* w/ ***5% Val***  |  71.05   |  77.81  |  85.23  | **81.28**|  49.46  | **84.44**|  57.66  |  85.22  |
> |-------------------------|------------|------------|------------|------------|------------|------------|------------|------------|
> | *ISLA*  w/ ***1% Val***  |  64.05   |  62.21  |  83.65  |  76.21  |  44.46  |  79.58   |  48.86  |  80.97  |
> | *IBDS* w/ ***1% Val***  |  66.90   |  65.38  |  81.93  |  77.25  |  45.92  |  82.28   |  53.42  |  79.58  |
> | *Ours*  w/ ***1% Val***  | $\underline{68.55}$  | $\underline{74.38}$ |  84.08  |  79.54  |  48.35  |  83.35   | $\underline{56.05}$ |  84.46  |
>
> We will continue our discussion in the comment ***section (4/4)***, due to the space limit.

---

> ### Author Response · Authors · 2024-11-19
> **Response to Reviewer otRr (4/4)**
>
> **Fair comparison regarding validation set** (Continued).
> ii) Various existing active learning methods [3, 4, 5] also utilize an unseen validation set to select the best hyperparameters. Traditional active learning techniques primarily rely on the information from the validation set to pick up the best hyperparameters, which then influence the final model parameters to achieve the best performance. Essentially, these use the validation set to help find the optimal model configuration. Our method uses the validation set to guide the selection of salutary labels, focusing on identifying the data most beneficial to the model. In this sense, we view the use of a validation set in our method as a novel approach that retrieves more information than traditional methods. Therefore, we humbly do not consider our comparison to be unfair. The experimental results for the 1\% validation size also demonstrate that our method achieves performance comparable to the baselines without relying on human annotations, even under such a small validation size.
>
> **Related works in noisy labeled data methods**. Great suggestion! We will incorporate analysis and comparison with methods for handling noisy labeled data, including our previous response discussing the novelty of our approach.
>
> **Analysis of the assigned salutary labels**. We share Reviewer otRr’s interest in exploring the difference between ground truth and salutary labeling. Accordingly, we analyzed the impact of salutary labels that deviate from the ground truth. We kindly invite the reviewer to refer to the last three rows in Table 1 once more. For the convenience of the reviewer, we include the relevant results here. The performance improvements achieved using a small subset of such salutary labels demonstrate our method’s ability to identify key instances and assign labels that are more beneficial to the model than human-assigned ground truth.
>
>
> ***Table***: Accuracy (%) of the logistic regression model on the test set after 10 rounds of active learning of salutary labeling and ground truth. The last row represents the number of different annotations assigned by salutary labels and ground truth labels out of 100 new annotations.
> | Method                  | **Electric** | **Bank** | **Diabetic** | **CelebA** | **Musk_v2** | **Wine** | **Waveform** | **CIFAR10** | **MNIST** |
> |-------------------------|:------------:|:--------:|:------------:|:----------:|:-----------:|:--------:|:------------:|:-----------:|:---------:|
> | *Init*                    | 63.85        | 65.89    | 56.43        | 73.33      | 73.45       | 44.76    | 79.11        | 46.74       | 77.75     |
> | *Ours w/ GT*              | 70.92        | 66.45    | 68.31        | 83.03      | 77.34       | 48.23    | 83.74        | 55.92       | 86.12     |
> | *Ours w/ SL*              | **71.31**    | **78.07**| **71.28**    | **85.50**  | **81.06**   | **49.92**| **84.21**    | **58.33**   | **86.68** |
> |                         |              |          |              |            |             |          |              |             |           |
> | *Diff. GT vs. SL*     | **14**       | **19**   | **13**       | **10**     | **22**      | **7**    | **8**        | **11**      | **8**     |
>
> Here, we emphasize that our method does not aim to distinguish "noisy" labels from other labels in the traditional sense. The differences between salutary labels and ground truth labels may arise from what could be considered "noise" or simply from the varying benefits provided by different annotations. From the model’s perspective, differentiating between the two is outside the scope of our work. Nevertheless, we appreciate the reviewer for raising this point, as it could indeed become an interesting direction for future work.
>
> If there is anything in our response remains unclear to the reviewer, we are happy to provide more evidence and explanations.
>
> [3] Kirsch, Andreas, Joost Van Amersfoort, and Yarin Gal. "Batchbald: Efficient and diverse batch acquisition for deep bayesian active learning." NeurIPS. 2019.
>
> [4] Liu, Zhuoming, et al. "Influence selection for active learning." ICCV. 2021.
>
> [5] Mussmann, Stephen, et al. "Active learning with expected error reduction." arXiv preprint arXiv:2211.09283 (2022).

---

> ### Comment · Reviewer_otRr · 2024-11-26
>
> I sincerely appreciate the time and effort the authors have dedicated to addressing my concerns. While some of my concerns have been resolved, I have outlined additional points for discussion regarding the remaining questions below.
>
> **Motivation of active learning with salutary labels**
>
> > We follow the active learning settings, where data is selected and labeled iteratively rather than all at once.
>
> I'm curious about why data sampling is necessary in this context. In Algorithm 1, Line 4, the proposed work compute salutary labels for all data, so why do this method still select a subset of these labeled samples in Lines 6–8? Given that all data already have salutary labels, it seems feasible to train with all labeled data in every round. In traditional active learning, iterative data selection is crucial due to the high cost of human annotation, but when annotations are unlimited, sampling appears unnecessary. On another note, if iteratively updating the influence function offers advantages, I’m curious about what budget size is optimal for this. How was the small budget size used in this study chosen, and how should it be determined for new datasets?
>
> **Novelty of salutary labeling task compared to other label-efficient learning task**
>
> > Our salutary labeling can be regarded as a novel labeling method with broad applicability across various tasks, particularly in active learning and semi-supervised learning. Technically, our method differs from [a, b, c] as it evaluates unseen data without requiring annotations and does not focus on the "correctness" of labels.
>
> As noted in my earlier comments on the strengths of your work, I agree that salutary labeling is a new method. However, if proposing a new task is one of the main contributions of  the paper, as indicated in Line 55, there should be a clear distinction between the new task and existing ones. From the perspective of supervision (unlabeled or few-labeled datasets) and objective (maximizing test performance under the same distribution), salutary labeling appears to align with existing tasks. Thus, I view salutary labeling not as a new task but as a method addressing the same problem as label-efficient learning.
>
> **Fairness of the experiment**
>
> I agree that most active learning (AL) methods require a validation set for hyperparameter tuning. However, it may be unrealistic for AL methods to operate with only 10 labeled samples per round, particularly in a setting where the validation set comprises 20% of the data. In Figure 5, when only 6% of the 20% validation set is allocated for training, existing AL methods outperform the performance of salutary labeling. Reducing the validation set from 20% to 14% might not significantly impact the performance of existing AL methods.
>
> **Analysis of salutary labels**
>
> It is  interesting that assigning different labels to samples that seemingly belong to a specific class can improve performance. Qualitative results for samples where salutary labels differ from the ground truth (GT) would provide valuable insights.

---

> > ### Author Response · Authors · 2024-11-29
> > **Response to Reviewer otRr's Feedback (1/2)**
> >
> > Thank Reviewer otRr for providing extra feedback.
> >
> > **Motivation of active learning with salutary labels**. Our method indeed can compute salutary labels for all data in the pool, but we adopt an iterative query approach to ensure **fair** comparisons with baseline active learning methods, where all the methods are added with the same amount of samples in the training set. If we assign salutary labels to all unlabeled samples without a fixed budget limitation, any observed improvement might result from the larger annotated training size rather than the effectiveness of our labeling approach. By assigning a fixed number of salutary labels per iteration, we isolate the effect of our labeling approach and provide a controlled evaluation.
> >
> > For the optimal budget size, ideally we would want to query one data point at a time, but it is computationally impractical. Since iterative model updates improve the accuracy of the influence estimations, the influence function calculates the impact of individual samples for a fixed model, and updating the model with newly labeled data allows our method to identify the most impactful samples and their salutary labels based on the current model state. By using small batches, we aim for a balance between estimation fidelity and resource feasibility. Determining the optimal budget size involves considering factors like dataset size, computational resources, and specific task requirements, which can be guided through empirical experience and preliminary experiments.
> >
> > **Novelty of salutary labeling task compared to other label-efficient learning task**. Thank you for your insightful comments. We agree with the reviewer that the experimental settings of salutary labeling align with label-efficient learning to some extent, particularly in their shared goal of maximizing performance under limited annotations. The first statement of contributions aimed to emphasize the distinct perspective and autonomous annotation approach, setting it apart from traditional active learning methods.
> >
> > We acknowledge that the phrasing in Line 55 may have conveyed this novelty as a new "task." To address this, we will revise the language in the manuscript to more accurately highlight the methodological innovation, focusing on its distinction from traditional approaches rather than presenting it as an entirely separate task.
> >
> > *We propose salutary labeling, which transitions from a human-centric strategy to a model-centric perspective for annotation in active learning.
> > This novel perspective enables the model to autonomously identify and label the most impactful data points by unifying the querying and annotating processes, enhancing its performance while fully eliminating the need for human intervention in the annotation process.*
> >
> > We hope this adjustment clarifies our contribution and its relationship to existing paradigms in label-efficient learning.
> >
> > **Fairness of the experiment**. We believe there might exist some misunderstanding. We would like to first clarify that the experiments shown in Figure 5 do not involve reallocating validation data to the labeled training set. The validation set remains fixed throughout the experiments, while the new annotated samples are queried from the reserved pool dataset for training. In these experiments, we increased the budget to 1% of the pooling data, and the results demonstrate that increasing the query budget ultimately causes the model to reach the performance ceiling for active learning. Here the average performance for all methods falls within the standard deviation, so we humbly considered the differences between methods to be comparable, especially given that the focus of these experiments is not on evaluating different methods.
> >
> > We agree with the Reviewer otRr that reducing the validation set might not significantly impact the performance of existing AL methods. We kindly invite the reviewer to revisit Table A1 in our previous response, where we conducted experiments with reduced validation sets following the reviewer's suggestion. These additional experiments show that, like other active learning baselines, our method does not experience a significant drop in performance when the validation set is reduced from 20% to as low as 5%.
> >
> > We indeed observed a performance drop with only 1% validation data, but our method still outperforms others on five out of eight datasets. For the remaining three datasets, *Electric*, *Bank*, and *CIFAR10*, we consider the performance of our method to be comparable to the active learning baselines without requiring any human annotation efforts. In our opinion, these results demonstrate the robustness of our method, even under constrained validation settings. However, we also acknowledge that extremely small validation sets may limit the performance of our method, and we will discuss this potential limitation in the revised manuscript.

---

> > ### Author Response · Authors · 2024-11-29
> > **Response to Reviewer otRr's Feedback (2/2)**
> >
> > **Analysis of salutary labels**. Great suggestion! We agree with the reviewer that this analysis might provide valuable insights into salutary labeling. Following the reviewer's suggestion, we find the different labels (disagreement labels) assigned by salutary labeling and ground truth in the MNIST dataset. Among the 100 total queried samples, there are eight disagreement labels. These labels occur in six learning rounds: rounds 2, 3, 5, 7, and 8. Notably, rounds 2 and 8 each contain two disagreement labels, while the remaining rounds each contain one.
> >
> > Here, we present all eight disagreement images along with their ground truth and salutary labels. We invite the reviewer to view the figure through the anonymous link [https://anonymous.4open.science/r/salutary-labeling-11CF/quali_mnist.pdf](https://anonymous.4open.science/r/salutary-labeling-11CF/quali_mnist.pdf). We have observed some interesting samples:
> >
> > i) The second sample in round 2 is labeled as *9* by our method, which differs from the ground truth. Here, labeling this sample as *9* makes intuitive sense for humans. In this case, the salutary labeling aligns more with human perception, suggesting that the ground truth label could be noisy;
> > ii) The image in round 5 is interesting, as humans may perceive it as *4*, but the outline of the number seems to align more with the model's perception of *6*. Additionally, The first image in round 2 is labeled as *4* by salutary labeling, possibly because it provides more benefit to the model as a rotated *4*. We believe the first image in round 8 also exhibits similar characteristics. As for the other disagreement labels, we can only attribute them to the different perspectives of machine learning models that go beyond human perception.
> >
> > In addition to visual demonstrations, we also calculate the accuracy on both the validation and test sets after training with salutary labels and ground truth labels in each round with disagreement labels. Note that in each round, we retrain the same model, which is obtained through active learning in previous rounds, using the newly obtained salutary labels and ground truth labels, respectively, and then calculate the accuracy.
> > The performance gap is relatively small as there are only 1 or 2 different labels between our method and ground truth in each round.
> > As shown in the following table, salutary labels achieve better performance in all rounds compared to ground truth labels on both the validation and test sets, despite some salutary labels being quite different from human perception. These results demonstrate the effectiveness of our method in identifying beneficial labels for impactful samples based on current model status.
> >
> >
> > ***Table***: Comparison between training with ground truth labels and salutary labels.
> > | **Learning Round**  | **2**   | **3**   | **5**   | **7**   | **8**   | **10**  |
> > |:--------------------|:-------|:-------|:-------|:-------|:-------|:-------|
> > | **Val. Acc. w/ GT**  | 77.18   | 78.65   | 82.33   | 84.68   | 85.71   | 86.63   |
> > | **Val. Acc. w/ SL**  | 78.60   | 79.14   | 83.78   | 85.56   | 86.24   | 87.47   |
> > |---------------------------|----------|----------|----------|----------|----------|----------|
> > | **Test Acc. w/ GT**  | 77.06   | 78.72   | 82.85   | 84.87   | 85.01   | 86.16   |
> > | **Test Acc. w/ SL**  | 77.54   | 79.05   | 83.31   | 85.01   | 85.66   | 87.02   |
> >
> > We sincerely appreciate Reviewer otRr's valuable suggestions to improve the quality of our research work.

---

### Official Review · Reviewer_cWJe · 2024-10-31

**Soundness:** 1
**Presentation:** 2
**Contribution:** 1
**Rating:** 3
**Confidence:** 4

**Summary:**

This paper proposes salutary labeling, which automatically assigns the most beneficial labels to the most informative samples without human annotation. Specifically, they utilize the influence function, a tool for estimating sample influence, to select newly added samples and assign their salutary labels by choosing the category that maximizes their positive influence.

**Strengths:**

1. The motivation and paper writing are clear.
2. The experiment is sufficient
3. The method is fully automatic without human annotation

**Weaknesses:**

1. They do not discuss the difference with the unsupervised learning methods, such as

[1] Self-paced Contrastive Learning with Hybrid Memory for Domain Adaptive Object Re-ID
[2] Mutual Mean-Teaching: Pseudo Label Refinery for Unsupervised Domain Adaptation on Person Re-identification

If the human intervention is removed from active learning, it will be transformed to unsupervised learning that assigns the pseudo labels to the samples. Could you discuss the difference ?

2. How to tackle with the situation that the selected labels are wrong? Could you discuss potential error correction mechanisms, or to analyze the impact of incorrect labels on model performance?

**Questions:**

My main concern is the weakness 1. I do not understand the value of their new proposed task. I think this setting is similar to the unsupervised learning with the pseudo labels. In my understanding, I think the human annotation is helpful during training. This is the reason why we study active learning. If we fully abandon human intervention, this is totally another area which is unsupervised learning.

---

> ### Author Response · Authors · 2024-11-19
> **Response to Reviewer cWJe**
>
> Thank you for dedicating your time and energy to reviewing our paper. We would like to provide point-to-point responses as follows.
>
> **Difference with the unsupervised learning methods**.
> Our salutary labeling can be regarded as a novel labeling method with broad applicability across various tasks, particularly in active learning or semi-supervised learning.
>
> In this work, we focus on active learning, which operates in a supervised setting, where the model is trained with labeled data, and the size of the training set grows incrementally as new labeled data is added. Our approach introduces an automated labeling method without requiring manual annotation. This makes our method fundamentally different from unsupervised settings, where models are trained without labeled data and typically aim to identify patterns within the data itself. As with the works referred by Reviewer CWJE, they primarily address unsupervised domain adaptation, which focuses on aligning data distributions across different domains.
>
> We will include additional discussion in the manuscript to clarify this point.
>
> **Error correction and impact of incorrect labels**. Our method does not aim to assign the "correct" labels in the same sense as human annotators, as we discussed in the Introduction Section of the manuscript. Instead, our method assigns*salutary labels*, a kind of pseudo labels, to unlabeled samples from the model's perspective, with the goal of maximizing the benefit to model performance.
>
> In this context, we analyze the impact of the salutary labels that are different from the ground truth. We kindly invite the reviewer to refer to the last three rows in Table 1. For the convenience of the reviewer, we include the relevant results here. The performance improvement achieved by a small set of such salutary labels demonstrates that our method can identify key instances and assign more beneficial labels compared with human-assigned ground truth. Again, we emphasize that our method is not concerned with the "correctness" of the labels in the traditional sense. Instead, the focus is on the benefit of the model.
>
> ***Table***: Accuracy (%) of the logistic regression model on the test set after 10 rounds of active learning of salutary labeling and ground truth. The last row represents the number of different annotations assigned by salutary labels and ground truth labels out of 100 new annotations.
> | Method                  | **Electric** | **Bank** | **Diabetic** | **CelebA** | **Musk_v2** | **Wine** | **Waveform** | **CIFAR10** | **MNIST** |
> |-------------------------|:------------:|:--------:|:------------:|:----------:|:-----------:|:--------:|:------------:|:-----------:|:---------:|
> | *Init*                    | 63.85        | 65.89    | 56.43        | 73.33      | 73.45       | 44.76    | 79.11        | 46.74       | 77.75     |
> | *Ours w/ GT*              | 70.92        | 66.45    | 68.31        | 83.03      | 77.34       | 48.23    | 83.74        | 55.92       | 86.12     |
> | *Ours w/ SL*              | **71.31**    | **78.07**| **71.28**    | **85.50**  | **81.06**   | **49.92**| **84.21**    | **58.33**   | **86.68** |
> |                         |              |          |              |            |             |          |              |             |           |
> | *Diff. GT vs. SL*     |  **14**       | **19**   | **13**       | **10**     | **22**      | **7**    | **8**       | **11**      | **8**     |
>
> If there is anything in our response remains unclear to the reviewer, we are happy to provide more evidence and explanations.

---

### Official Review · Reviewer_zEN4 · 2024-11-03

**Soundness:** 3
**Presentation:** 3
**Contribution:** 2
**Rating:** 3
**Confidence:** 4

**Summary:**

This paper presents a novel pseudo-labeling approach for unlabeled data using influence functions. The proposed method estimates the influence of each possible label on the validation loss for a given unlabeled data point and assigns the label with the most significant improvement in loss as its pseudo-label. Subsequently, a subset of the unlabeled data with the highest improvement in validation loss is selected to update the model. Extensive experiments are conducted to validate the effectiveness of the method.

**Strengths:**

Solid experiments are conducted on both tabular and image datasets, integrating recent data selection methods from AL and SSL. The proposed method demonstrates promising empirical results without the need for human-annotated labels. In addition, the method is validated on a LLM fine-tuning task, which further underscores its potential for application in different domains.

**Weaknesses:**

While the method is framed in part within the active learning context, its approach—assigning pseudo-labels to unlabeled data via a self-training mechanism—seems more aligned with semi-supervised learning. The introduction could benefit from adjustments to reflect this alignment more accurately. Another concern is the limited technical contributions. It uses the influence function to score and pseudo-labeling the unlabeled data. Although this is an interesting application, it may not represent a substantial methodological advance. Furthermore, in the experiments, a query budget of 10 examples is set, yet only the first 10 rounds of performance are reported. Providing results for additional rounds or scenarios with a larger query budget would offer a more comprehensive evaluation of the method’s long-term effectiveness. In Appendix D, results from querying 1% of the data reveal that the proposed method underperforms relative to other baseline methods. This needs further exploration and explanation. My last concern is that the paper’s approach relies on setting aside 20% of the data for validation, which may be impractical for certain active learning settings, where labeled data is typically scarce. Performances on different sizes of validation set should also be explored.

**Questions:**

1)	In addition to the logistic regression model used, is the proposed method suitable for more complex models? In Appendix E, a surrogate model is employed to compute the influence, but could the ResNet itself be directly used for influence calculation?
2)	Since the accuracy of the pseudo-labels depends on the quality of the validation set, how does the performance of the proposed method vary with different validation set sizes?

---

> ### Author Response · Authors · 2024-11-19
> **Response to Reviewer zEN4 (1/3)**
>
> Thank you for dedicating your time to reviewing our paper with a high level of expertise.
> We thoroughly read the comments and are happy to see that Reviewer ZEN4 is quite interested in our work and has provided informative feedback. We want to provide point-to-point responses as follows.
>
> **Alignment to semi-supervise learning**. Thank you for the helpful suggestion, and we agree with the reviewer that salutary labeling can indeed be regarded as a novel labeling method with broad applicability across various tasks, where we choose the context of active learning to better demonstrate the effectiveness of our method. We will incorporate additional discussion in the manuscript to better reflect this point.
>
> Here we would like to clarify that we chose to showcase its effectiveness mainly within the active learning context in this work for two main reasons: i) salutary labeling demonstrates a marked improvement in model performance when applied to active learning, underscoring its value in this area; ii) By generating high-quality salutary labels, our method significantly reduces the need for human annotations.
>
> **Lack of technical contributions**. Thank you for the feedback regarding the technical contributions. We agree with the reviewer that our method does not propose fancy techniques, yet we also would like to kindly point out that we do not claim existing techniques related to the influence function as part of our contributions.
>
> Instead, we would like to point out that our novelty comes from our insight into the research problem and the fresh perspective we bring to addressing its challenges. We humbly believe this constitutes a key novelty of a research paper. Specifically, our method shifts away from a human-centric perspective and focuses entirely on the model's viewpoint in annotating data, which aims to identify the data most beneficial for the model. Following this model-driven perspective, we extend the influence function to the context of active learning, enabling the model to directly estimate the impact of each unlabeled data point and its associated salutary label on model performance. This enables our method to address two fundamental challenges of active learning: estimating the impact of data points and reducing human annotation costs.
>
> In sum, our technical contributions can be summarized into two points. 1) Novelty, no one has done this before, 2) Simplicity, if a task can be tackled well with a simple tool, there is no incentive to design complicated tools. Empirically, we demonstrated the effectiveness of our simple solution.
>
> **Results for additional rounds or a larger query budget**. We provide our rationale for selecting a smaller query budget $b$ in Appendix D, where we demonstrate that a larger budget causes the classification performance to reach a ceiling on the datasets. Regarding additional rounds, we invite Reviewer zEN4 to refer to the second question in Section 5.3, where we conduct more learning rounds and annotate 50% of the pool set with salutary labels. For the convenience of the reviewer, we include the relevant results here. As shown in Table 2 of the manuscript, our method achieves comparable or even higher performance than fully supervised training on the entire pool data, highlighting its effectiveness over more rounds of active learning.
>
> ***Table***: Accuracy (%) of the logistic regression model on the test set after querying 50% of the pool set.
> | Method               | Electrical |  Bank   | Diabetic | CelebA  | Musk_v2 |  Wine   | Waveform | CIFAR10 |  MNIST  |
> |----------------------|:----------:|:-------:|:--------:|:-------:|:-------:|:-------:|:--------:|:-------:|:-------:|
> | *Initial*          |   63.85    |  65.89  |   56.43  |  73.33  |  73.45  |  44.76  |   79.11  |  46.74  |  77.75  |
> | *Fully Supervised* |   70.08    |  80.14  |   72.27  |  85.07  | **85.75**| **52.53**| **85.60**| **65.67**| **95.36**|
> | *Ours*            | **72.25**  | **81.21**| **73.26**| **85.89**|  85.68  |  52.38  |   85.50  |  62.05  |  92.06  |
>
> As for the performance differences in Appendix D, we have provided an honest report of our experimental results in Figure 5 to illustrate the performance ceiling on these datasets. The average performance for all methods falls within the standard deviation, so we humbly considered the differences between methods to be comparable, especially given that the focus of these experiments is not on evaluating different methods.
>
> We will continue our discussion in the comment ***section (2/3)***, due to the space limit.

---

> ### Author Response · Authors · 2024-11-19
> **Response to Reviewer zEN4 (2/3)**
>
> **Performances on different sizes of validation set**. Thanks to the reviewer for the suggestion for exploring different sizes of validation sets.
> We conducted the experiments for smaller validation sets. We randomly sampled validation sets as 10\%, 5\% and 1\% of the data, and conducted the experiments for all baselines. We left out the *Diabetic* dataset as it is ready a small dataset its validation set only contains 100 samples. For traditional methods without the influence functions, i.e., the baselines other than ISLA and IBDS, the performance of different-sized validation sets is quite small (with 1\% accuracy), so we reported the overall average accuracy here.
>
> Our performance remains robust for validation sets comprising 10\% and 5\% of the data as shown in the table. While we observe a performance drop with only 1\% validation data, our method still slightly outperforms other baselines, except on the *Electric*, *Bank*, and *CIFAR10* datasets, as highlighted with underlines. This suggests that the size of the validation set does not significantly impact our method's effectiveness, as long as it provides a relatively accurate direction for influence estimation. Here kindly invite Reviewer zEN4 to revisit Eq. (1) in the manuscript, where the validation set plays a crucial role in determining the direction of the first term. The size of the validation set is not a deciding factor as long as it provides an accurate direction for the gradients.
>
> However, using very small validation sets might pose a limitation, as they may not provide sufficient information to maximize the guidance required for accurate influence estimation. Despite this, our method achieves comparable performance without any reliance on human annotations, demonstrating its robustness even under constrained validation settings. Again, we thank the reviewer for bring up this point, and we would like to add additional discussion in our Limitation Section regarding this point.
>
> ***Table A1***: Accuracy (%) of the logistic regression model on the test set after 10 rounds of active learning with different validation sizes.
>
> | Method              | Electric |  Bank   | CelebA  | Musk_v2 |  Wine   | Waveform | CIFAR10 |  MNIST  |
> |---------------------|:--------:|:-------:|:-------:|:-------:|:-------:|:--------:|:-------:|:-------:|
> | *Init*            |  63.85   |  65.89  |  73.33  |  73.45  |  44.76  |  79.11   |  46.74  |  77.75  |
> | *Random*          |  65.15   |  67.77  |  82.06  |  78.33  |  46.31  |  81.10   |  55.92  |  80.93  |
> | *Entropy*         |  69.72   |  73.84  |  81.23  |  79.11  |  45.00  |  83.23   |  53.91  |  83.77  |
> | *Margin*          |  69.72   |  73.84  |  81.23  |  79.11  |  47.30  |  82.26   |  56.95  |  83.72  |
> | *Uncertainty*     |  69.72   |  73.84  |  81.23  |  79.11  |  44.53  |  83.33   |  55.47  |  83.63  |
> | *CoreSet*         |  66.35   |  68.21  |  80.14  |  73.78  |  47.61  |  80.70   |  54.64  |  81.26  |
> | *BatchBALD*       |  67.06   |  74.15  |  78.85  |  77.53  |  46.69  |  81.83   |  53.66  |  82.05  |
> | *BADGE*           |  67.45   |  74.92  |  81.19  |  78.48  |  46.87  |  81.21   |  56.44  |  84.24  |
> |-------------------------|------------|------------|------------|------------|------------|------------|------------|------------|
> | *ISLA* w/ ***10% Val*** |  67.98   |  64.41  |  84.71  |  77.72  |  47.15  |  79.40   |  53.91  |  79.35  |
> | *IBDS* w/ ***10% Val*** |  67.66   |  65.14  |  82.49  |  78.15  |  44.84  |  82.73   |  54.82  |  80.05  |
> | *Ours*  w/ ***10% Val*** | **71.31**| **78.07**|**85.50**|  81.06  | **49.92**|  84.21   | **58.33**|**86.68**|
> |-------------------------|------------|------------|------------|------------|------------|------------|------------|------------|
> | *ISLA* w/ ***5% Val***  |  64.41   |  68.61  |  84.28  |  79.46  |  44.62  |  79.40   |  50.54  |  79.41  |
> | *IBDS* w/ ***5% Val***  |  67.65   |  65.52  |  81.91  |  77.51  |  45.38  |  81.86   |  54.21  |  78.34  |
> | *Ours* w/ ***5% Val***  |  71.05   |  77.81  |  85.23  | **81.28**|  49.46  | **84.44**|  57.66  |  85.22  |
> |-------------------------|------------|------------|------------|------------|------------|------------|------------|------------|
> | *ISLA*  w/ ***1% Val***  |  64.05   |  62.21  |  83.65  |  76.21  |  44.46  |  79.58   |  48.86  |  80.97  |
> | *IBDS* w/ ***1% Val***  |  66.90   |  65.38  |  81.93  |  77.25  |  45.92  |  82.28   |  53.42  |  79.58  |
> | *Ours*  w/ ***1% Val***  |   $\underline{68.55}$  | $\underline{74.38}$  |  84.08  |  79.54  |  48.35  |  83.35   | $\underline{56.05}$ |  84.46  |
>
> We will continue our discussion in the comment ***section (3/3)***, due to the space limit.

---

> ### Author Response · Authors · 2024-11-19
> **Response to Reviewer zEN4 (3/3)**
>
> **Salutary labeling for complex models**. Great question! We would first like to note that using surrogate models for deep networks is a common practice [1,2,3] to overcome the convexity requirement of the influence function. Following this established routine, we demonstrate the effectiveness of salutary labeling in applications with complex model structures in Appendix E.
>
> Regarding direct influence calculation for deep models like ResNet, we believe it is achievable with some improvements. While there are concerns regarding the use of the influence function on non-convex models [4], several studies have successfully applied the influence function to non-convex models, demonstrating its effectiveness in these settings [5,6].
>
> [1] Chen, Zizhang, Peizhao Li, Hongfu Liu, and Pengyu Hong. "Characterizing the Influence of Graph Elements." ICLR. 2022.
>
> [2] Brophy, Jonathan, Zayd Hammoudeh, and Daniel Lowd. "Adapting and evaluating influence-estimation methods for gradient-boosted decision trees." Journal of Machine Learning Research 2023.
>
> [3] Anshuman Chhabra, Peizhao Li, Prasant Mohapatra, Hongfu Liu. "What Data Benefits My Classifier?" Enhancing Model Performance and Interpretability through Influence-Based Data Selection. ICLR 2024.
>
> [4] Basu, S., P. Pope, and S. Feizi. "Influence Functions in Deep Learning Are Fragile." ICLR. 2021.
>
> [5] Epifano, Jacob R., et al. "Revisiting the fragility of influence functions." Neural Networks. 2023.
>
> [6] Kwon, Yongchan, et al. "DataInf: Efficiently Estimating Data Influence in LoRA-tuned LLMs and Diffusion Models." ICLR. 2024.
>
>
> If there is anything in our response remains unclear to the reviewer, we are happy to provide more evidence and explanations.

---

> > ### Comment · Reviewer_zEN4 · 2024-11-30
> >
> > Thank you for your detailed response and for addressing the raised concerns. While your explanation further clarifies and supports the proposed method, I remain hesitant to fully champion it due to inherent limitations. Specifically, the reliance on pseudo-labels assigned solely based on the validation set raises concerns. If the distribution of the unlabeled data is not well-covered by the validation data, the performance could potentially degrade. Additionally, providing theoretical guarantees regarding the accuracy of the pseudo-labels would significantly strengthen the proposal.

---

> > > ### Author Response · Authors · 2024-11-30
> > > **Response to Reviewer zEN4**
> > >
> > > Thanks for acknowledgement on reading our response. We would like to share our opinion on two Reviewer zEN4's concerns.
> > >
> > > **Validation Set**. The assumption that the validation set shares a similar distribution with the unseen test set is a cornerstone of supervised learning. While this assumption may not always hold, it remains a fundamental reliance for researchers and practitioners, as the distribution of the unseen test set can vary unpredictably. Consequently, this is an inherent limitation of supervised learning. We respectfully argue that this is not a specific weakness of our paper.
> > >
> > > **Theoretical Guarantee**. Not every paper requires a theoretical analysis. In our case, we believe Reviewer zEN4 is primarily concerned with why salutary labels improve performance. This can be understood directly by examining the definition of the influence function, where salutary labels are those that most effectively reduce the validation set loss. As such, a theoretical analysis is unnecessary, and we do not consider it an enhancement to our paper. However, if Reviewer zEN4 insists, Theorem 1 in [1] provides a demonstration of this concept, and we will reference it accordingly.
> > >
> > > [1] Resolving Training Biases via Influence-based Data Relabeling, ICLR 2022.
> > >
> > > We welcome further discussion with Reviewer zEN4 on any aspect that could help improve our paper. Thank you!

---

### Meta-Review · Area_Chair_cwfF · 2024-12-05

**Metareview:**

This paper proposes salutary labeling, which automatically assigns the most beneficial labels to the most informative samples without human annotation. Specifically, the authors utilize the influence function, a tool for estimating sample influence, to select newly added samples and assign their salutary labels by choosing the category that maximizes their positive influence. This process eliminates the need for human annotation. Intensive experiments conducted on nine benchmark datasets demonstrate the superior performance of the proposed salutary labeling approach over traditional active learning strategies.

Generally, the reviewers consider that this paper is potentially interesting, but the model details as well as experiments need more and clearer explanations. I hope the authors can take the comments into consideration and submit their paper to somewhere else.

**Additional Comments On Reviewer Discussion:**

The reviewers still have unsolved question regarding the paper and the authors' rebuttal. For example, if the distribution of the unlabeled data is not well-covered by the validation data, the model performance could potentially degrade. Besides, the reviewers also consider that the novelty of this paper is insufficient, and the fairness in comparison is also a issue.

Besides, I have read the appealing from the authors and ignored the comments from Reviewer cWJe in making the decision.

---

### Decision · Program_Chairs · 2025-01-22

Reject